# Hyperparameter Trajectory Inference with Conditional Lagrangian Optimal Transport

**Harry Amad, Mihaela van der Schaar**
Department of Applied Mathematics and Theoretical Physics
University of Cambridge, UK
`hmka3@cam.ac.uk`

## Abstract

Neural networks (NNs) often have critical behavioural trade-offs that are set at design time with hyperparameters—such as reward weights in reinforcement learning or quantile targets in regression. Post-deployment, however, user preferences can evolve, making initial settings undesirable, necessitating potentially expensive retraining. To circumvent this, we introduce the task of Hyperparameter Trajectory Inference (HTI): to learn, from observed data, how a NN's conditional output distribution changes with its hyperparameters, and construct a surrogate model that approximates the NN at unobserved hyperparameter settings. HTI requires extending existing trajectory inference approaches to incorporate conditions, exacerbating the challenge of ensuring inferred paths are feasible. We propose an approach based on conditional Lagrangian optimal transport, jointly learning the Lagrangian function governing hyperparameter-induced dynamics along with the associated optimal transport maps and geodesics between observed marginals, which form the surrogate model. We incorporate inductive biases based on the manifold hypothesis and least-action principles into the learned Lagrangian, improving surrogate model feasibility. We empirically demonstrate that our approach reconstructs NN outputs across various hyperparameter spectra better than other alternatives.

## 1 Introduction

Neural network (NN) behaviour is critically shaped by hyperparameters, $\lambda$, which alter the parameters of the trained network, $\theta_\lambda$, thereby affecting the distribution of outputs $y$ given input $x$, $p_{\theta_\lambda}(y|x)$.[1] Often, hyperparameters govern subjective trade-offs, requiring users to fix complex preferences at design time. In deployment, however, evolving conditions can render initial hyperparameters suboptimal, necessitating retraining, which can be infeasible. This motivates an alternate approach—to learn a surrogate model that can sample outputs across a spectrum of hyperparameter settings. We introduce **Hyperparameter Trajectory Inference (HTI)**—inspired by trajectory inference (TI) (Hashimoto et al., 2016; Lavenant et al., 2021)—to address exactly this. The goal of HTI is to learn hyperparameter-induced dynamics $\lambda \mapsto p_{\theta_\lambda}(y|x)$ between observed distributions $\{p_{\theta_\lambda}(y|x)\}_{\lambda \in \Lambda_{\mathrm{obs}}}$ and develop a surrogate model $\hat{p}(y|x,\lambda)$ with which the NN conditional probability paths, for some reasonable hyperparameters $\lambda \in \Lambda$, can be estimated as $(\hat{p}(y|x,\lambda))_{\lambda \in \Lambda} \approx (p_{\theta_\lambda}(y|x))_{\lambda \in \Lambda}$, permitting approximate inference-time adjustment of $\lambda$. Below we expand on two potential use cases of HTI.

> **Reinforcement learning.** NN-based RL policies (Zhu et al., 2023; Park et al., 2025) define a state-conditional action distribution $p_{\theta_\lambda}(a|s)$, with fundamental behaviours determined by certain hyperparameters. Consider, for instance, a policy for cancer treatment, with a reward function balancing two objectives: reducing tumour volume, and minimising immune system damage, weighted by a scalar $\lambda$. The ideal balance can vary per patient, based on factors such as comorbidities (Sarfati et al., 2016). An HTI surrogate policy $\hat{p}(a|s,\lambda)$ would allow for personalised treatment strategies, by varying $\lambda$ at inference time (§5.2.1).

---

[1] $p_{\theta_\lambda}(y|x) = \delta_{\theta_\lambda(x)}(y)$ in the deterministic case, but we also consider other distributions e.g. generative models, or distributions parameterised by NN outputs.

> **Quantile regression.** Regression tasks can require measures of uncertainty. Quantile regression (Koenker & Bassett Jr, 1978) provides a way to construct prediction intervals, but typically models target individual quantiles $\tau$, or a multi-head model outputs a fixed set of quantiles (Wen et al., 2018). This can make examining arbitrary quantiles, to tailor uncertainty bounds, computationally intensive. HTI can address this, learning the dynamics $\tau \mapsto p_{\theta_\tau}(y|x)$ across a desired quantile range, yielding a surrogate that can predict all intermediate quantiles (§5.3).

HTI is challenging, as the dynamics $\lambda \mapsto p_{\theta_\lambda}(y|x)$ are typically non-linear, given complex deep learning optimisation landscapes (Ly & Gong, 2025), making simple interpolation schemes, like conditional flow matching (CFM) (Lipman et al., 2023; Liu et al., 2023; Albergo & Vanden-Eijnden, 2023), unlikely to yield feasible surrogates $(\hat{p}(y|x, \lambda))_{\lambda \in \Lambda}$. HTI requires an approach capable of capturing complex, non-Euclidean dynamics from sparse ground-truth distributions. Similar problems have been addressed in standard TI (Tong et al., 2020; Scarvelis & Solomon, 2023; Kapusniak et al., 2024; Pooladian et al., 2024), however the effects of *conditions* on probability paths, which is essential for HTI, are currently under-explored.

We aim to enable HTI by addressing this problem of conditional TI (CTI). We propose an approach grounded in conditional Lagrangian optimal transport (CLOT) theory, allowing us to bias inferred conditional probability paths to remain meaningful. Specifically, we aim to learn kinetic and potential energy terms that define a Lagrangian cost function, and to encode inductive biases into these terms. This cost function determines what is deemed to be efficient movement between $\{p_{\theta_\lambda}(y|x)\}_{\lambda \in \Lambda_{\text{obs}}}$, and we use neural approximate solutions to the optimal transport maps and geodesics that respect this Lagrangian cost to infer conditional probability paths. We do so in a manner inspired by Pooladian et al. (2024), extending this method to handle conditions, encode more useful inductive biases, and perform on more complex and higher-dimensional geometries. Once the Lagrangian and CLOT components are learned, samples for a target hyperparameter $\lambda_{\text{target}} \in \Lambda$ and condition $x$ can be drawn by sampling from a base distribution in $\{p_{\theta_\lambda}(y|x)\}_{\lambda \in \Lambda_{\text{obs}}}$, approximating CLOT maps and geodesic paths, and evaluating the paths at the $\lambda_{\text{target}}$ position. In short, our main contributions include:

1. We introduce the problem of **Hyperparameter Trajectory Inference** to enable inference-time NN behavioural adjustment, using the framing of TI to encourage particular inductive biases for modelling hyperparameter dynamics (§2.1).
2. We propose a general method for CTI to learn complex conditional dynamics from temporally sparse ground-truth samples, based on principles from CLOT (§4). We extend the procedure of Pooladian et al. (2024) in several ways, learning a data-dependent potential energy term $\mathcal{U}$ alongside a kinetic term $\mathcal{K}$ (§4.1), elevating the method to the conditional OT setting (§4.2), and establishing a more expressive neural representation for the learned metric, $G_{\theta_G}$, underpinning $\mathcal{K}$ that naturally extends to higher dimensions (§4.3).
3. We demonstrate empirically that our approach reconstructs conditional probability paths better than alternatives in multiple applications of HTI, enabling effective inference-time adaptation of a single hyperparameter in various domains (§5).

## 2 PRELIMINARIES

### 2.1 HYPERPARAMETER TRAJECTORY INFERENCE

TI (Hashimoto et al., 2016; Lavenant et al., 2021) aims to recover the continuous time-dynamics $t \mapsto p_t$ of a population from observed samples from a set of temporally sparse distributions $\{p_t\}_{t \in \mathcal{T}_{\text{obs}}}$. CTI is an extension of TI where a conditioning variable $x \in \mathcal{X}$ affects these dynamics, with a goal of inferring the conditional population dynamics $t \mapsto p_t(\cdot|x)$ for arbitrary $x$.

Building upon the concept of CTI, we introduce a novel instantiation that we address in this work—HTI. In HTI, the 'population' is the outputs of a NN, with distribution $p_{\theta_\lambda}(y|x)$ conditioned on its input $x$, and we wish to learn the dynamics $\lambda \mapsto p_{\theta_\lambda}(y|x)$ induced by a single continuous hyperparameter $\lambda \in \Lambda$ (acting as 'time') from a set of known distributions $\{p_{\theta_\lambda}\}_{\lambda \in \Lambda_{\text{obs}}}$ to recover the conditional probability paths $(p_{\theta_\lambda}(y|x))_{\lambda \in \Lambda}$. These dynamics can be used to build a surrogate model $\hat{p}(y|x, \lambda)$ for the NN in question, allowing efficient, approximate sampling at arbitrary hyperparameter settings within $\Lambda$. Since many hyperparameters, by virtue of their effects during NN training, define families of NNs among which the optimal member is context dependent, such a surrogate model could reduce the need to retrain NNs in dynamic deployment scenarios.

## 2.2 Conditional optimal transport

We deploy the framework of conditional optimal transport (COT) (Villani, 2008) to define optimal maps and paths between conditional distributions, which can be neurally approximated. Let $\mathcal{Y}_0$ and $\mathcal{Y}_1$ be two complete, separable metric spaces, and $\mathcal{X}$ be a general conditioning space. For $x \in \mathcal{X}$, consider probability measures $\mu_0(\cdot|x) \in \mathcal{P}(\mathcal{Y}_0)$ and $\mu_1(\cdot|x) \in \mathcal{P}(\mathcal{Y}_1)$ and cost function $c(\cdot, \cdot|x) : \mathcal{Y}_0 \times \mathcal{Y}_1 \to \mathbb{R}_{\geq 0}$. The primal COT formulation (Kantorovich, 1942) involves a coupling $\pi$ that minimises the transport cost:

$$\text{COT}_c(\mu_0(\cdot|x), \mu_1(\cdot|x)) = \inf_{\pi(\cdot, \cdot|x) \in \Pi(\mu_0(\cdot|x), \mu_1(\cdot|x))} \int_{\mathcal{Y}_0 \times \mathcal{Y}_1} c(y_0, y_1|x) d\pi(y_0, y_1|x) \tag{1}$$

where $\Pi(\mu_0(\cdot|x), \mu_1(\cdot|x))$ is the collection of all probability measures on $\mathcal{Y}_0 \times \mathcal{Y}_1$ with marginals $\mu_0(\cdot|x)$ on $\mathcal{Y}_0$ and $\mu_1(\cdot|x)$ on $\mathcal{Y}_1$. Solving this primal problem is generally intractable, and it cannot be easily neurally approximated as it requires modelling a high-dimensional joint distribution. The equivalent dual formulation simplifies the problem to a constrained optimisation over two scalar potential functions $f(\cdot|x)$ and $g(\cdot|x)$:

$$\text{COT}_c(\mu_0(\cdot|x), \mu_1(\cdot|x)) = \sup_{f,g} \int_{\mathcal{Y}_0} f(y_0|x) d\mu_0(y_0|x) + \int_{\mathcal{Y}_1} g(y_1|x) d\mu_1(y_1|x) \tag{2}$$

subject to the constraint $f(y_0|x) + g(y_1|x) \leq c(y_0, y_1|x), \forall(y_0, y_1) \in (\mathcal{Y}_0, \mathcal{Y}_1)$. Enforcing this constraint with neural instantiations of $f$ and $g$ across the entire domain is challenging (Seguy et al., 2018). As such, we follow recent literature (Makkuva et al., 2020; Amos, 2023; Pooladian et al., 2024) and utilise the semi-dual formulation based on the $c$-transform, converting the problem into an unconstrained optimisation over a single potential $g(\cdot|x)$:

$$\text{COT}_c(\mu_0(\cdot|x), \mu_1(\cdot|x)) = \sup_g \int_{\mathcal{Y}_0} g^c(y_0|x) d\mu_0(y_0|x) + \int_{\mathcal{Y}_1} g(y_1|x) d\mu_1(y_1|x) \tag{3}$$

where $g^c(\cdot|x)$ is the $c$-transform of $g(\cdot|x)$:

$$g^c(y_0|x) := \inf_{y_1' \in \mathcal{Y}_1} \{c(y_0, y_1'|x) - g(y_1'|x)\}. \tag{4}$$

Denoting $g^*(\cdot|x)$ as an optimal potential for (3), the COT map $T_c(\cdot|x) : \mathcal{Y}_0 \to \mathcal{Y}_1$ can be found as

$$T_c(y_0|x) \in \underset{y_1' \in \mathcal{Y}_1}{\text{argmin}} \{c(y_0, y_1'|x) - g^*(y_1'|x)\}. \tag{5}$$

## 2.3 Conditional Lagrangian optimal transport

In the above, the cost function $c$ is where knowledge of system dynamics can be embedded, to shape the COT maps and paths (Asadulaev et al., 2024). The standard Euclidean cost $c(y_0, y_1) = \|y_0 - y_1\|^2$, for example, deems straight paths as the most efficient. To induce more complex paths, given our assumed complex hyperparameter-dynamics, we require a cost function that is path-dependent, which motivates us to use principles from Lagrangian dynamics (Goldstein et al., 1980), bringing us to the CLOT setting. Given a smooth, time-dependent curve $q_t$ for $t \in [0, 1]$, with time derivative $\dot{q}_t$, and a Lagrangian $\mathcal{L}(q_t, \dot{q}_t|x)$, the *action* of $q$, $\mathcal{S}(q|x)$, can be determined as

$$\mathcal{S}(q|x) = \int_0^1 \mathcal{L}(q_t, \dot{q}_t|x) dt. \tag{6}$$

The resulting Lagrangian cost function $c$ can then be defined using the *least-action*, or *geodesic*, curve between $y_0$ and $y_1$

$$c(y_0, y_1|x) = \inf_{q:q_0=y_0, q_1=y_1} \mathcal{S}(q|x). \tag{7}$$

We denote geodesics as $q^*$. While flexible in form, a common Lagrangian instantiation is

$$\mathcal{L}(q_t, \dot{q}_t|x) = \mathcal{K}(q_t, \dot{q}_t|x) - \mathcal{U}(q_t|x) = \frac{1}{2}\dot{q}_t^T G(q_t|x)\dot{q}_t - \mathcal{U}(q_t|x) \tag{8}$$

where $\mathcal{K}$ and $\mathcal{U}$ are kinetic and potential energy terms, respectively, with metric $G$ defining the geometry of the underlying manifold (e.g. for Euclidean manifolds, $G = I$). We consider *learning* conditional Lagrangians of the above form by setting a neural representation of $G$ and estimating $\mathcal{U}$ using a kernel density estimate, and learning neural estimates of the transport maps and geodesics for the consequent CLOT problem. We design $\mathcal{U}$ and $G$ to incorporate biases for dense traversal and least-action into the inferred conditional probability paths.

## 3   RELATED WORKS

**Trajectory inference.** TI (Hashimoto et al., 2016; Lavenant et al., 2021) is prominent in domains such as single-cell genomics, where destructive measurements preclude tracking individual cells over time (Macosko et al., 2015; Schiebinger et al., 2019). Successful TI relies on leveraging inductive biases to generalise beyond the sparse observed times. One typical bias is based on least-action principles—assuming that populations move between observed marginals in the most efficient way possible—naturally giving rise to OT approaches (Yang & Uhler, 2019; Schiebinger et al., 2019; Tong et al., 2020; Scarvelis & Solomon, 2023; Pooladian et al., 2024). Another potential bias invokes the manifold hypothesis (Bengio et al., 2013), which posits that data resides on a low-dimensional manifold, concentrated around the observed data (Arvanitidis et al., 2022; Chadebec & Allassonnière, 2022), to encourage inferred paths to traverse dense regions of the data space (Kapusniak et al., 2024).

**Neural optimal transport.** NNs have been used for OT, especially in high-dimensions where classical OT algorithms are infeasible (Makkuva et al., 2020; Korotin et al., 2021). The semi-dual OT formulation with neural parametrisations of the Kantorovich potentials and transport maps is standard (Makkuva et al., 2020; Amos, 2023; Pooladian et al., 2024). Neural COT has also been explored (Wang et al., 2024; 2025), although with *fixed* cost functions, and we novelly extend this to incorporate learned conditional Lagrangian costs. Our work is particularly related to Scarvelis & Solomon (2023) and Pooladian et al. (2024), which jointly learn OT cost functions and resulting transport maps from observed time marginals. We consider using more expressive forms for the cost function, involving Lagrangians with kinetic and potential energy terms, and we operate in the conditional OT setting.

**Conditional generative modeling via density transport.** Some generative models, such as conditional diffusion (Ho & Salimans, 2022) and CFM models (Zheng et al., 2023), operate by transporting mass from a source to a target distribution, according to some condition. They can therefore be applied to conditional TI. However, generative models focus on accurately learning the target data distribution, and they are generally unconcerned with the intermediate distributions formed along the transport paths. While some recent works utilise OT principles to achieve more efficient learning and sampling for CFM models (Tong et al., 2024; Pooladian et al., 2023), their primary objective remains high-fidelity sample generation from the target distribution.

**Bayesian optimization.** Bayesian hyperparameter optimization (Snoek et al., 2012; Shahriari et al., 2015) builds a surrogate model of a NN's objective function across hyperparameters. HTI extends this by learning a surrogate for the NN's conditional output distribution rather than for a scalar objective function. HTI could allow for more flexible Bayesian hyperparameter optimisation with arbitrary, post-hoc objective functions calculated with surrogate samples (Appendix A).

## 4   NEURAL CONDITIONAL LAGRANGIAN OPTIMAL TRANSPORT

We now present our method for general CTI, which involves a neural approach to CLOT. From observed temporal marginals, we seek to learn both the underlying conditional Lagrangian $\mathcal{L}(q, \dot{q}|x) = \mathcal{K}(q, \dot{q}|x) - \mathcal{U}(q|x)$ that governs dynamics, along with the consequent CLOT maps $T_c$ and geodesics $q^*$, such that conditional trajectories can be inferred. We novelly encode both the inductive biases discussed in Section 3—least-action and dense traversal—into $\mathcal{L}$ to aid generalisation of inferred trajectories beyond the observed temporal regions.

### 4.1   POTENTIAL ENERGY TERM

Firstly, we set the conditional potential energy, $\hat{\mathcal{U}}(q|x)$, through which we encode a bias for dense traversal. By designing $\hat{\mathcal{U}}(q|x)$ to be large in dense regions of the data space, and small elsewhere, the Lagrangian cost function $c$, as in (7), will lead to geodesics that favour dense regions.

Let $\mathcal{D}_{obs} = \{(y_i, x_i, t_i)\}_{i=1}^N$ be the set of observed samples, where $y_i \in \mathcal{Y}$ are the $D_y$-dimensional ambient space observations, $x_i \in \mathcal{X}$ are their $D_x$-dimensional conditions, and $t_i \in \{t_0, t_1, ..., t_T\}$ are the $T$ 'times' of observation. We define the potential at $q \in \mathcal{Y}$ for a given condition $x \in \mathcal{X}$ as:

$$\hat{\mathcal{U}}(q|x) = \alpha \, \log(\hat{p}(q|x) + \epsilon), \tag{9}$$

where $\alpha > 0$ is set by the user to control the strength of the density bias, $\epsilon > 0$ is for numerical stability, and $\hat{p}(q|x)$ is estimated with a Nadaraya-Watson estimator (Nadaraya, 1964; Watson, 1964):

$$\hat{p}(q|x) = \frac{\sum_{i=1}^{N} K_{h_y}(q - y_i) K_{h_x}(x - x_i)}{\sum_{j=1}^{N} K_{h_x}(x - x_j)}, \tag{10}$$

where $K_{h_y}$ and $K_{h_x}$ are Gaussian kernel functions with bandwidths $h_y$ and $h_x$, respectively:

$$K_{h_y}(u) = (2\pi h_y^2)^{-D_y/2} \exp\left(-\frac{||u||^2}{2h_y^2}\right), \; K_{h_x}(v) = (2\pi h_x^2)^{-D_x/2} \exp\left(-\frac{||v||^2}{2h_x^2}\right). \tag{11}$$

We can see that (9) will be high when $\hat{p}(q|x)$ is high, and low when $\hat{p}(q|x)$ is low, thus encoding our desired bias for geodesics to traverse dense regions of the data space. $\hat{\mathcal{U}}(q|x)$ is fixed throughout the subsequent learning phase for the kinetic energy term $\mathcal{K}$ and the CLOT maps and geodesic paths.

## 4.2 JOINT LEARNING OF KINETIC ENERGY TERM AND CLOT PATHS

To learn the remaining kinetic term $\mathcal{K}(q, \dot{q}|x) = \frac{1}{2}\dot{q}^T G(q|x)\dot{q}$, and solve the consequent CLOT problem, we adopt a neural approach similar to Pooladian et al. (2024), adapting it to our conditional setting. We operate under the assumption that the observed data display dynamics that are efficient in the underlying data manifold, to embed the desired least-action bias into our method. We consider neural instantiations of the metric $G_{\theta_G}$ within $\mathcal{K}$, and the $T$ Kantorovich potentials $g_{\theta_{g,k}}$ defining the CLOT problems between temporally adjacent observed distributions, with parameters $\theta_G$ and $\theta_{g,k}$ respectively.[2] These networks are learnt with a min-max procedure, alternating between optimising $G_{\theta_G}$, with fixed $g_{\theta_{g,k}}$, to minimise the estimated CLOT cost between observed marginals (encoding the desired least-action principles), and optimising each $g_{\theta_{g,k}}$, with fixed $G_{\theta_G}$, to maximise (3) (to accurately estimate the CLOT cost under the current metric). The overall objective is

$$\min_{\theta_G} \sum_k \mathbb{E}_x \left[ \max_{\theta_{g,k}} \mathbb{E}_{y_k \sim \mu_k(\cdot|x)}[g_{\theta_{g,k}}^c(y_k|x)] + \mathbb{E}_{y_{k+1} \sim \mu_{k+1}(\cdot|x)}[g_{\theta_{g,k}}(y_{k+1}|x)] \right], \tag{12}$$

where $\mu_k(\cdot|x)$ is the conditional distribution of the data at time $t_k$. We denote the inner maximisation objective for each interval as $\mathcal{L}_{\text{dual}}^{(k)}(\theta_{g,k})$, and the outer minimisation objective as $\mathcal{L}_{\text{metric}}(\theta_G)$.

Calculating $g^c$, as in (4), requires solving an optimisation problem, with a further embedded optimisation problem to calculate the transport cost. These nested optimisations can make training computationally infeasible. As such, we adopt the amortisation procedure from Pooladian et al. (2024), simultaneously training and using neural approximators to output CLOT maps $T_{\theta_{T,k}}(y_k|x) \approx T_{c,k}(y_k|x)$ and the parameters of a spline-based geodesic estimation, $q_\varphi \approx q^*$, allowing efficient $c$-transform approximation. At a given training iteration, the current learned map $T_{\theta_{T,k}}$ warm-starts the minimisation (4); this estimate is refined with a limited number of L-BFGS (Liu & Nocedal, 1989) steps to yield $T_{c,k}(y_k|x)$ which is used to calculcate $g^c$ in (12), and as a regression target for $T_{\theta_{T,k}}$:

$$\mathcal{L}_{\text{map}}(\theta_{T,k}) = \mathbb{E}\left[(T_{\theta_{T,k}}(y_k|x) - T_{c,k}(y_k|x))^2\right]. \tag{13}$$

To efficiently calculate the cost function required for these L-BFGS steps we approximate geodesic paths $q^*$ with a cubic spline $q_\varphi$, with parameters $\varphi$ output by a NN $S_{\theta_S}$ trained to minimise

$$\mathcal{L}_{\text{path}}(\theta_S) = \mathbb{E}\left[\mathcal{S}(q_\varphi|x)\right], \; \varphi = S_{\theta_S}(y_k, T_{\theta_{T,k}}(y_k|x), x). \tag{14}$$

To condition each network on $x$, we equip them with FiLM layers (Perez et al., 2018) that modulate the first-layer activations based on $x$. The overall training procedure (Algorithm 1) alternates between updating each $\theta_{g,k}$, $\theta_{T,k}$, and $\theta_s$ to maximise the inner part of (12), minimise (13), and minimise (14), respectively, and updating $\theta_G$ to minimise the outer sum in (12).

## 4.3 METRIC PARAMETRISATION

Within the above procedure, the parametrisation of the neural metric $G_{\theta_G}$ is particularly important, as this must be a symmetric, positive-definite, $D_y$-dimensional matrix to be a valid metric. Critically,

---

[2] $g_{\theta_{g,k}}$ denotes the $k$th Kantorovich potential, for the CLOT between the distributions at $t_k$ and $t_{k+1}$

---

**Algorithm 1** Neural CLOT Training

---

**Require:** Observed data $\mathcal{D}_{\text{obs}}$, ambient and conditional bandwidths $h_y, h_x$, potential weight $\alpha$, no. outer training iterations $N_{\text{outer}}$, no. inner training iterations $N_{\text{inner}}$, learning rates $\eta_g, \eta_T, \eta_S, \eta_G$

1: $\hat{\mathcal{U}}(q|x) \leftarrow \alpha \log(\hat{p}(q|x))$, where $\hat{p}(q|x) = \frac{\sum_{i=1}^{N} K_{h_y}(q-y_i)K_{h_x}(x-x_i)}{\sum_{j=1}^{N} K_{h_x}(x-x_j)}$

2: Initialise $\theta_G, \{\theta_{g,k}, \theta_{T,k}\}_k, \theta_S$

3: Define $\mathcal{S}(q|x) := \int_0^1 (\frac{1}{2}\dot{q}_t^T G_{\theta_G}(q_t|x)\dot{q}_t - \hat{\mathcal{U}}(q_t|x))dt$

4: **for** $i = 1 \ldots N_{\text{outer}}$ **do**

5:     **for** $j = 1 \ldots N_{\text{inner}}$ **do**

6:         **for** $k = 0 \ldots T-1$ **do**

7:             $\mathcal{D}_k \leftarrow \{(y, x, t) \in \mathcal{D}_{\text{obs}} \mid t = t_k\}$

8:             **for** $(y_k, x) \in \mathcal{D}_k$ **do**

9:                 $y_k' \leftarrow T_{\theta_{T,k}}(y_k|x)$

10:                $y_k'^* \leftarrow \text{L-BFGS}(y_k', \mathcal{S}(q_\phi|x) - g_{\theta_{g,k}}(y_k'|x))$, where $\phi = S_{\theta_S}(y_k, y_k', x)$

11:                $g_{\theta_{g,k}}^c(y_k|x) \leftarrow \mathcal{S}(q_{\phi^*}|x) - g_{\theta_{g,k}}(y_k'^*|x)$, where $\phi^* = S_{\theta_S}(y_k, y_k'^*, x)$

12:         **end for**

13:         $\theta_{g,k} \leftarrow \theta_{g,k} + \eta_g \nabla \mathcal{L}_{\text{dual}}^{(k)}(\theta_{g,k}), \theta_{T,k} \leftarrow \theta_{T,k} - \eta_T \nabla \mathcal{L}_{\text{map}}(\theta_{T,k})$

14:         **end for**

15:     $\theta_S \leftarrow \theta_S - \eta_S \nabla \mathcal{L}_{\text{path}}(\theta_S)$

16:     **end for**

17:     $\theta_G \leftarrow \theta_G - \eta_G \nabla \mathcal{L}_{\text{metric}}(\theta_G)$

18: **end for**

19: **return** $\{T_{\theta_{T,k}}\}_k, S_{\theta_S}$

---

there exist degenerate minima to (12) by setting $G_{\theta_G} \to \mathbf{0}$, where movement in all directions results in near-zero cost. We set our parametrisation to ensure $G_{\theta_G}$ avoids this and maintains sufficient volume. In Pooladian et al. (2024), where only two-dimensional data spaces are considered, they set $G_{\theta_G}$ as a fixed diagonal matrix with a neural rotation matrix

$$G_{\theta_G}(x) = \begin{bmatrix} \cos(R_{\theta_G}(x)) & -\sin(R_{\theta_G}(x)) \\ \sin(R_{\theta_G}(x)) & \cos(R_{\theta_G}(x)) \end{bmatrix} \begin{bmatrix} 1 & 0 \\ 0 & 0.1 \end{bmatrix} \begin{bmatrix} \cos(R_{\theta_G}(x)) & -\sin(R_{\theta_G}(x)) \\ \sin(R_{\theta_G}(x)) & \cos(R_{\theta_G}(x)) \end{bmatrix}^T \quad (15)$$

where $R_{\theta_G}(x)$ is the output of the NN. This is only applicable to two-dimensional spaces, and avoids degenerate solutions by *fixing* the local anisotropy of $G_{\theta_G}$. We design a parametrisation that extends to higher dimensions, and is more expressive, while still avoiding degenerate solutions without requiring regularisation as in Scarvelis & Solomon (2023). Specifically, we set $G_{\theta_G}$ using its eigendecomposition $G_{\theta_G} = R_{\theta_G} E_{\theta_G} R_{\theta_G}^T$, where a NN parametrises *both* a $D_y$-dimensional diagonal matrix $E_{\theta_G}$, and rotation matrices $R_{\theta_G}$. To avoid degeneracy, we enforce the entries of $E_{\theta_G}$, and therefore the eigenvalues of $G_{\theta_G}$, to be positive, and sum to a non-zero 'eigenvalue budget', ensuring non-trivial volume of $G_{\theta_G}$ while permitting expressive levels of anisotropy. To define the $D_y$-dimensional rotation matrix $R_{\theta_G}$, we multiply $\frac{D_y(D_y-1)}{2}$ Givens rotation matrices (Givens, 1958), with angles parametrised by the NN. This can improve performance over the fixed approach of Pooladian et al. (2024) in two-dimensions (§5.5), while also extending to higher dimensions (§5.3).

### 4.4 SAMPLING ALONG THE INFERRED TRAJECTORY

To generate samples from the inferred conditional distribution $\hat{p}(y|x, t^*)$, we use the neural approximators for the CLOT maps and geodesics, avoiding the need for any optimisation at inference time. First, samples are drawn from the ground-truth distribution with the largest observed base time with $t_k < t^*$, $y_k \sim p_{t_k}(\cdot|x)$. The learned map $T_{\theta_{T,k}}(y_k|x)$ then predicts the transported point $y_{k+1}$ at the end of the interval $[t_k, t_{k+1}]$, which contains $t^*$. Subsequently, the parameters for the approximate geodesic path $q_\varphi$ connecting $y_k$ to $y_{k+1}$ can be estimated as $\varphi = S_{\theta_s}(y_k, y_{k+1}, x)$, and $q_\varphi$ can be evaluated at the appropriate time. By normalising $t^*$ to $s^* = (t^* - t_k)/(t_{k+1} - t_k)$, the inferred sample is obtained as $\hat{y}_{t^*} = q_\varphi(s^*)$.

## 5 EXPERIMENTS

We now empirically demonstrate the efficacy of our method for CTI and some specific HTI scenarios. All results are averaged over 20 runs, and reported with standard errors. We provide detailed experimental set-ups in Appendix C.

### 5.1 ILLUSTRATIVE EXAMPLE OF CTI

To illustrate our method's inductive biases, we devise a temporal process with conditions $x \in \{1, 2, 3, 4\}$, where each defines a distribution $p_t(y|x)$ that evolves from the origin over $t \in [0, 1]$ as a noised von Mises, with centre moving along one of four semicircular paths. Samples from the true process are shown in Figure 1, where each condition has a distinct colour, and lighter samples are from larger $t$. To conduct CTI, using observations from $t \in \{0, 0.5, 1.0\}$, models must: (1) learn condition-dependent dynamics despite overlapping initial distributions; (2) capture the non-Euclidean geometry of semicircular paths; and (3) generalise across $t \in [0, 1]$ from sparse temporal samples.

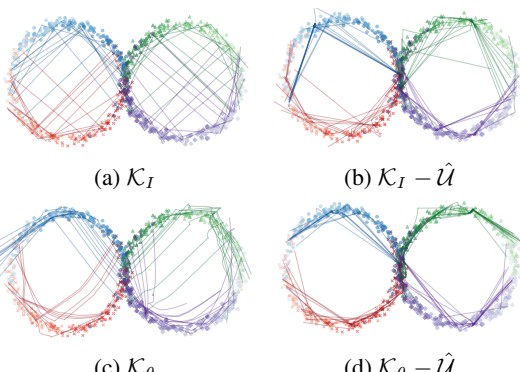

(a) $\mathcal{K}_I$  (b) $\mathcal{K}_I - \hat{\mathcal{U}}$

(c) $\mathcal{K}_\theta$  (d) $\mathcal{K}_\theta - \hat{\mathcal{U}}$

Figure 1: Dots represent true samples from the temporal process across $t \in [0, 1]$, lines represent model estimated trajectories from $t = 0$ to $t = 1$. Each condition has a distinct colour.

We compare four ablations of our method, with varying complexity of the learned conditional Lagrangian: (1) $\mathcal{K}_I$: Using an identity metric $G = I$ and setting $\hat{\mathcal{U}} = 0$, resulting in Euclidean geometry with no density bias; (2) $\mathcal{K}_\theta$: Learning the metric $G_{\theta_G}$ via our method in §4.2 and setting $\hat{\mathcal{U}} = 0$, to incorporate only the inductive bias of least-action; (3) $\mathcal{K}_I - \hat{\mathcal{U}}$: Using an identity metric $G = I$ and learning $\hat{\mathcal{U}}$ as in §4.1, to incorporate only the inductive bias of dense traversal, and; (4) $\mathcal{K}_\theta - \hat{\mathcal{U}}$: Our full approach, learning *both* the metric $G_{\theta_G}$ and the potential term $\hat{\mathcal{U}}$.

Figure 1 shows the inferred paths of samples from $t = 0$ to $t = 1$. Our full method (Figure 1d) most faithfully reconstructs the true temporal process, as the paths correctly diverge according to their condition and closely follow the intended semicircular geometry. We can see the individual effects of both inductive biases, as individually learning $\hat{\mathcal{U}}$ (Figure 1b) results in straight paths that favour denser regions, avoiding the circle centres, while learning

Table 1: NLL and CD at $t \in \{0.25, 0.75\}$.

| Method | NLL ($\downarrow$) | CD ($\downarrow$) |
|--------|--------|--------|
| $\mathcal{K}_I$ | 105.713 (2.42) | 0.323 (0.003) |
| $\mathcal{K}_\theta$ | 23.008 (4.62) | 0.158 (0.009) |
| $\mathcal{K}_I - \hat{\mathcal{U}}$ | −0.532 (0.057) | 0.016 (0.001) |
| $\mathcal{K}_\theta - \hat{\mathcal{U}}$ | −0.662 (0.046) | 0.016 (0.001) |

$G_{\theta_G}$ only (Figure 1c) better captures the underlying curvature of the semicircular geometry. In Table 1 we evaluate $\hat{p}(y|x, t)$ at withheld $t \in \{0.25, 0.75\}$, reporting negative log-likelihood (NLL) and distance from the target circle perimeter (CD). We can quantitatively see that both inductive biases improve the feasibility of the inferred marginals.

### 5.2 HTI FOR REWARD-WEIGHTING IN REINFORCEMENT LEARNING

We now transition to specific applications of HTI, first addressing a compelling challenge in RL, to create surrogate policies that allow for dynamic reward weighting.

#### 5.2.1 CANCER THERAPY

We investigate HTI for personalised cancer therapy, mirroring the first use case presented in §1. We employ an environment from `DTR-Bench` (Luo et al., 2024), which we call `Cancer`, that simulates tumour progression under chemotherapy and radiotherapy. Natural Killer (NK) cells are pivotal immune system components, and they can be depleted as a side effect of cytotoxic treatments like chemotherapy and radiotherapy (Shaver et al., 2021; Toffoli et al., 2021), increasing susceptibility to infections and compromising treatment efficacy. This side effect varies substantially with age,

Table 2: Average surrogate `Cancer` reward across $\lambda_{\text{nk}} \in \{1, 2, 3, 4, 6, 7, 8, 9\}$.

| Method | Reward ($\uparrow$) |
|---|---|
| Direct | $-38.35$ (10.65) |
| NLOT | $9.26$ (10.55) |
| $\mathcal{K}_\theta$ | $30.63$ (8.50) |
| CFM | $36.03$ (6.46) |
| MFM | $41.05$ (4.16) |
| $\mathcal{K}_I$ | $48.72$ (7.22) |
| $\mathcal{K}_I - \hat{\mathcal{U}}$ | $83.62$ (5.37) |
| $\mathcal{K}_\theta - \hat{\mathcal{U}}$ | $102.49$ (5.46) |

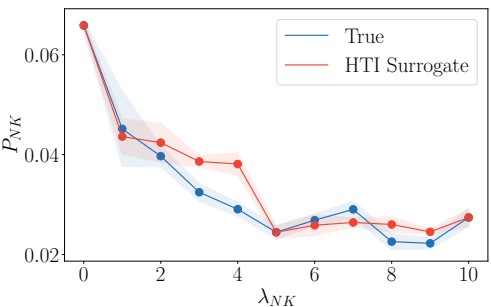

Figure 2: $P_{NK}$ vs. $\lambda_{\text{nk}}$ for ground truth policies and our surrogate policy.

comorbidities, and baseline immune status (Diakos et al., 2014) and, consequently, optimal cancer therapy necessitates a patient-specific balance between tumour reduction and NK cell preservation.

The `Cancer` reward function incorporates both tumour volume and NK cell preservation, with a hyperparameter $\lambda_{\text{nk}}$ weighting an NK cell penalty term, $P_{NK}$. Training a Proximal Policy Optimization (PPO) (Schulman et al., 2017) agent to convergence in this environment takes approximately 3.5 hours, so training per-patient policies with tailored $\lambda_{\text{nk}}$ is computationally prohibitive. This therefore presents a prime application for HTI, to enable inference-time policy adaptation.

To learn the $\lambda_{\text{nk}}$-induced dynamics of the policy distribution across $\lambda_{\text{nk}} \in [0, 10]$, we train ground-truth policies with PPO at $\lambda_{\text{nk}} \in \{0, 5, 10\}$ and sample 1000 state-action pairs from each converged policy, across a shared set of states, to act as the HTI training set. We assess the four approaches from §5.1, alongside some baselines. We compare to: (1) a direct surrogate, where the target $\lambda_{\text{nk}}$, current state, and action from $\lambda_{\text{nk}} = 0$ are inputs to an MLP that is trained to output actions at a given $\lambda_{\text{nk}}$ via supervised regression; (2) a CFM surrogate (Lipman et al., 2023), which learns a vector field between the distributions at $\lambda_{\text{nk}} \in \{0, 5, 10\}$ and generates samples by integrating actions to the desired $\lambda_{\text{nk}}$ point; (3) a metric flow matching (MFM) surrogate (Kapusniak et al., 2024) that is similar to CFM, but biases the vector field to point towards dense regions of the data space; and (4) the NLOT method of Pooladian et al. (2024). We add FiLM conditioning to these to make them appropriate for HTI.

In Table 2 we report the average reward for each surrogate at held-out $\lambda_{\text{nk}} \in \{1, 2, 3, 4, 6, 7, 8, 9\}$. Our full method ($\mathcal{K}_\theta - \hat{\mathcal{U}}$) infers the most realistic trajectory between settings, yielding a surrogate policy with the best average reward. We also examine NK cell preservation in Figure 2, plotting the average per-episode $P_{NK}$ penalty for our surrogate and ground-truth policies. Our method closely mirrors the behaviour of the ground-truth policies, correctly favouring treatment that preserves NK cells as $\lambda_{\text{nk}}$ increases. Critically, training our surrogate model takes approximately 15 minutes, after which rapid inference-time adaptation is possible. This contrasts with the 3.5 hours required to train each new PPO policy, highlighting the substantial computational advantage conferred by HTI.

### 5.2.2 REACHER

To further demonstrate HTI for reward weighting, we evaluate it in the `Reacher` environment from OpenAI Gym (Brockman et al., 2016), a standard continuous control benchmark. In this setting, an agent controls a two-joint arm with the goal of reaching a random target position. The reward is designed to minimise distance to the target, while penalising the magnitude of the joint torques, discouraging high-force movements, and it is weighted by a hyperparameter $\lambda_{\text{c}}$.

Similar to the cancer therapy experiment, we train PPO agents at $\lambda_{\text{c}} \in \{1, 5\}$, and collect 1000 state-action pairs from each agent to form the HTI training dataset. In Table 3 we evaluate the same suite of surrogate models as previously, assessing inferred policy

Table 3: Average surrogate `Reacher` rewards across $\lambda_{\text{c}} \in \{2, 3, 4\}$.

| Method | Reward ($\uparrow$) |
|---|---|
| Direct | $-6.711$ (0.070) |
| MFM | $-6.561$ (0.053) |
| $\mathcal{K}_I - \hat{\mathcal{U}}$ | $-6.397$ (0.031) |
| $\mathcal{K}_I$ | $-6.307$ (0.041) |
| CFM | $-6.251$ (0.028) |
| NLOT | $-6.173$ (0.038) |
| $\mathcal{K}_\theta$ | $-6.158$ (0.033) |
| $\mathcal{K}_\theta - \hat{\mathcal{U}}$ | $-6.093$ (0.036) |

behaviour at unseen $\lambda_c \in \{2, 3, 4\}$. Our full method ($\mathcal{K}_\theta - \hat{\mathcal{U}}$) again yields the most performant surrogate, achieving the highest average reward.

### 5.2.3 NON-LINEAR REWARD SCALARIZATION

The previous reward scalarizations involve linear combinations of a main objective (tumour volume/distance to target) and a penalty term (NK penalty/torque penalty). Such scalarization is known to lead to well-behaved trade-offs when tuning reward weights (Rădulescu et al., 2020). For a more challenging RL setting, with less well-behaved hyperparameter dynamics, we modify `Cancer` to have non-linear reward scalarization, with a hinge penalty. In this `Cancer_nl` setup, the weighted NK penalty is only applied if the change in cell count crosses a threshold (see definition in Appendix C.2.2). We employ the same training and evaluation protocol as in §5.2.1, with results in Table 4. We see that our method again achieves the highest average reward across held-out settings, remaining robust when the hyperparameter governs non-linear objectives.

Table 4: Average surrogate `Cancer_nl` reward across $\lambda_{nk} \in \{1, 2, 3, 4, 6, 7, 8, 9\}$.

| Method | Reward ($\uparrow$) |
|---|---|
| $\mathcal{K}_I$ | 42.84 (5.86) |
| Direct | 49.50 (17.90) |
| NLOT | 52.63 (6.52) |
| $\mathcal{K}_\theta$ | 54.44 (7.66) |
| CFM | 69.70 (7.73) |
| MFM | 78.44 (4.47) |
| $\mathcal{K}_I - \hat{\mathcal{U}}$ | 86.21 (6.32) |
| $\mathcal{K}_\theta - \hat{\mathcal{U}}$ | 91.94 (11.46) |

### 5.3 HTI FOR QUANTILE REGRESSION

We now demonstrate HTI's application in a higher-dimensional setting of quantile regression for time-series forecasting, mirroring the second use case presented in §1. Time-series forecasting is a task where providing a full picture of uncertainty, such as through quantile regression, is crucial, but the need to train forecasting models to target distinct quantiles can hinder this. We investigate whether HTI can address this by inferring intermediate quantiles from the outputs of models trained at the extremes of the quantile range. Using the ETTm2 forecasting dataset (Zhou et al., 2021), we train two MLPs to forecast a 3-step horizon from a 12-step history at the quantiles $\tau = 0.01$ and $\tau = 0.99$, using a standard pinball loss. We then generate a dataset of 1200 forecasts from these two models, across shared inputs, to act as the HTI training set. In Table 5 we evaluate the mean squared error (MSE) for surrogate forecasts at held-out quantiles $\tau \in \{0.1, 0.25, 0.5, 0.75, 0.9\}$ compared to the true NN outputs on unseen input data. Our full method once again outperforms all baselines. Figure 3 provides qualitative validation of this, visualising the central $80\%$ prediction intervals (between the $\tau = 0.1$ and $\tau = 0.9$ quantiles) from different surrogates on a random selection of samples, alongside the $80\%$ intervals from the ground-truth NNs. Our method most closely matches the width and shape of the true interval.

Table 5: MSE of surrogate ETTm2 forecasts compared to NNs trained across quantiles $\tau \in \{0.1, 0.25, 0.5, 0.75, 0.9\}$.

| Method | MSE ($\downarrow$) |
|---|---|
| Direct | 1.845 (0.065) |
| CFM | 1.402 (0.008) |
| MFM | 1.387 (0.022) |
| $\mathcal{K}_I$ | 0.765 (0.070) |
| $\mathcal{K}_I - \hat{\mathcal{U}}$ | 0.651 (0.076) |
| $\mathcal{K}_\theta$ | 0.620 (0.057) |
| $\mathcal{K}_\theta - \hat{\mathcal{U}}$ | 0.608 (0.034) |

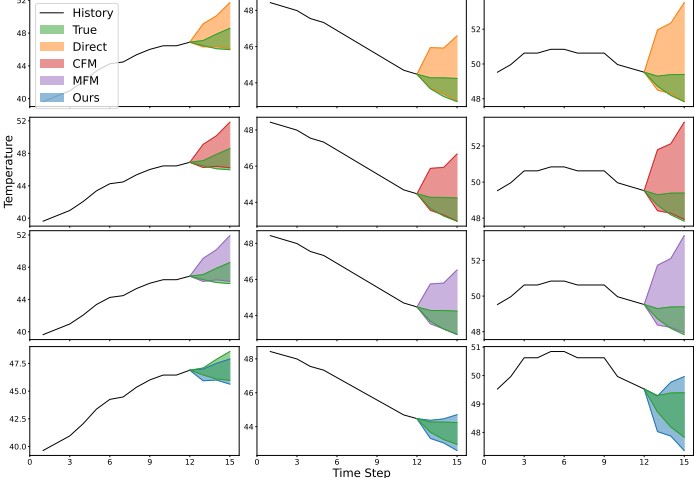

Figure 3: Central $80\%$ prediction intervals from HTI surrogates compared with the true intervals on randomly selected ETTm2 samples, for direct (top), CFM (second row), MFM (third row), and our (bottom) approach.

## 5.4 HTI FOR DROPOUT IN GENERATIVE MODELLING

To demonstrate a more general use case for HTI, we consider the dropout hyperparameter, which is often used for regularisation in NN training. We consider a simple setting, training a diffusion model (Ho et al., 2020) on the `sklearn` two moons dataset at dropout settings $p \in \{0, 0.5, 0.99\}$, and we draw 1000 samples from each model to act as the HTI training set. In Table 6 we evaluate surrogate models at held-out settings $p \in \{0.1, 0.2, 0.3, 0.4, 0.6, 0.7, 0.8, 0.9\}$, measuring the Wasserstein distance (WD) between surrogate and ground-truth distributions. Our methods incorporating the density bias $\hat{\mathcal{U}}$ perform well here, with $\mathcal{K}_I - \hat{\mathcal{U}}$ achieving the lowest WD. We see that HTI can be used to interpolate across a general hyperparameter such as dropout with minimal error in this setting.

Table 6: WD of surrogate two moons distributions from ground-truth diffusion models, trained at dropout $p \in \{0.1, 0.2, 0.3, 0.4, 0.6, 0.7, 0.8, 0.9\}$.

| Method | WD ($\downarrow$) |
|---|---|
| $\mathcal{K}_I$ | 0.570 (0.020) |
| MFM | 0.480 (0.020) |
| CFM | 0.464 (0.021) |
| Direct | 0.450 (0.013) |
| $\mathcal{K}_\theta$ | 0.404 (0.018) |
| NLOT | 0.291 (0.022) |
| $\mathcal{K}_\theta - \hat{\mathcal{U}}$ | 0.079 (0.003) |
| $\mathcal{K}_I - \hat{\mathcal{U}}$ | 0.060 (0.001) |

## 5.5 METRIC LEARNING ABLATION

In Table 7 we compare our neural metric $G_{\theta_G}$, with learned rotation $R_{\theta_G}$ and eigenvalues $E_{\theta_G}$, against the parametrisation from Pooladian et al. (2024), which uses fixed eigenvalues $E$. We evaluate both within our most expressive Lagrangian setting across the previous experiments with two-dimensional ambient spaces. Our parametrisation leads to better performance in most tasks, yielding a lower NLL in the semicircle task and higher rewards in the `Cancer` and `Reacher` environments. On the other hand, it achieves a slightly worse WD in the generative modelling dropout experiment. These results suggest that it is possible to learn the eigenvalues of $G_{\theta_G}$, and this can enable a more accurate recovery of the underlying conditional dynamics. Furthermore, unlike the parametrisation of Pooladian et al. (2024), ours readily extends to higher-dimensional settings, as demonstrated in §5.3.

Table 7: $G_{\theta_G}$ ablations in 2D experiments.

| | Semicircle | | Cancer | Reacher | Dropout |
|---|---|---|---|---|---|
| $G_{\theta_G}$ | NLL ($\downarrow$) | CD ($\downarrow$) | Reward ($\uparrow$) | Reward ($\uparrow$) | WD ($\downarrow$) |
| $R_{\theta_G} E R_{\theta_G}^T$ | $-0.602$ (0.033) | 0.016 (0.001) | 98.72 (6.32) | $-6.122$ (0.080) | 0.076 (0.003) |
| $R_{\theta_G} E_{\theta_G} R_{\theta_G}^T$ | $-0.662$ (0.046) | 0.016 (0.001) | 102.49 (5.46) | $-6.093$ (0.036) | 0.079 (0.003) |

## 6 DISCUSSION

In this work, we investigate CTI, proposing a novel methodology grounded in the principles of CLOT. Our approach extends existing TI techniques by explicitly incorporating conditional information, and novelly combining dense traversal (via $\hat{\mathcal{U}}$) and least-action (via $\mathcal{K}_\theta$) inductive biases. Our empirical results show we can effectively reconstruct non-Euclidean conditional probability paths from sparsely observed marginal distributions (§5.1). Our ablation study validates our neural metric parametrisation, highlighting its ability to capture intricate data geometries (§5.5) while extending to higher dimensions (§5.3). We also investigate performance at different sparsity levels in Appendix E. Furthermore, we propose HTI as a novel and impactful instantiation of CTI, addressing the challenge of adapting NN behaviour without expensive retraining. We showcased the practical utility of HTI for interpolating between reward weights in RL (§5.2), quantile targets in time-series forecasting (§5.3), and dropout settings in generative modelling (§5.4). For reference on the efficiency HTI can confer, the ground-truth result in Figure 2 required training 11 PPO policies, taking approximately 38 GPU hours, while the surrogate result requires training three PPO policies and an HTI surrogate, taking only 11 GPU hours. Further potential applications of HTI are discussed in Appendix A.

Nevertheless, HTI will be challenging when the underlying dynamics are chaotic, making inference from sparse samples inherently difficult. Given the relatively simple settings we demonstrate here, further investigation across a wider range of hyperparameter landscapes is warranted. Also, our method for HTI is only applicable for varying a single, continuous hyperparameter. Future work should explore extensions to handle multiple hyperparameters, which we discuss in Appendix D.

## REPRODUCIBILITY STATEMENT

We are committed to ensuring our work is reproducible. As such, we give a brief introduction to the mathematical concepts our method is based on in §2, clearly describe our method in §4, and provide concrete training and sampling algorithms in Algorithms 1 and 2, respectively. For the results in §5, we give detailed experimental set-ups in Appendix C. This includes detailing the datasets and environments used, model hyperparameters and training procedures, and providing references and links to key libraries. Furthermore, we release our code base here: `https://github.com/harrya32/hyperparameter-trajectory-inference`.

## ACKNOWLEDGEMENTS

Harry Amad's studentship is funded by Canon Inc. This work was supported by Azure sponsorship credits granted by Microsoft's AI for Good Research Lab.

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

# A  FURTHER APPLICATIONS OF HTI

We will now elaborate on some especially compelling potential applications for HTI. In general, HTI can be useful in scenarios when a user is deploying a NN in a dynamic environment, where behavioural preferences are context-dependent, and where the NN has a hyperparameter with a known, tangible behavioural effect. Traditionally, in such deployment scenarios, a user would either have to compromise with some fixed NN behavioural setting, determined at training time, or allow dynamic behaviours by undergoing slow and expensive retraining at different hyperparameter settings when deemed necessary. HTI can alleviate this by enabling much faster inference time NN behavioural adaptation, by sampling estimated outcomes from the surrogate model $\hat{p}(y|x, \lambda)$ for a novel $\lambda$ setting. For a visual depiction of HTI in action, see Figure 4.

## A.1  VARYING NEURAL NETWORK ROBUSTNESS IN DYNAMIC NOISE SETTINGS

Perturbations (e.g. Gaussian noise) of magnitude $\epsilon$ added to NN training data can increase robustness during inference for noisy inputs (Goodfellow et al., 2015; Madry et al., 2018). Calibrating the training noise to that expected to be seen in deployment can lead to optimal results in terms of inference-time accuracy. The hyperparameter $\epsilon$ directly controls this trade-off: higher $\epsilon$ typically increases robustness to noisy inputs but may decrease accuracy on clean data.

Consider an image classification NN used in a quality control system on a manufacturing line, where the input $x_{\text{image}}$ is an image of a product. The desired level of robustness $\epsilon^*$ might change based on several factors:

- **Environmental conditions:** Changes in factory lighting can alter image noise.
- **Operational mode:** A user might decide to temporarily increase sensitivity to minor defects (requiring lower $\epsilon^*$ for higher accuracy on subtle features) during a specific batch run, or prioritise overall stability (higher $\epsilon^*$) if the line is known to be experiencing vibrations.
- **Sensor age:** As the camera ages, its noise profile might change, warranting an adjustment to $\epsilon^*$.

HTI would learn a surrogate model $p(y_{\text{class}}|x_{\text{image}}, \epsilon)$. At inference time, based on the current conditions and any explicit user preference for robustness, an appropriate $\epsilon^*$ can be selected. The system then samples from $p(y_{\text{class}}|x_{\text{image}}, \epsilon^*)$ to obtain predictions as if from a model specifically tuned for that desired robustness level, without needing on-the-fly retraining.

## A.2  VARYING SHORT- VS. LONG-TERM FOCUS IN REINFORCEMENT LEARNING

The discount factor $\gamma \in [0, 1)$ in reinforcement learning (RL) determines an agent's preference for immediate versus future rewards. A low $\gamma$ leads to myopic, short-term reward-seeking behaviour, while a $\gamma$ closer to 1 encourages far-sighted planning, valuing future rewards more highly.

Consider an RL agent managing a patient's chronic disease treatment, such as Type 2 Diabetes, where actions involve adjusting medication dosage or recommending lifestyle interventions. The state $s$ includes physiological markers (e.g., blood glucose levels, HbA1c) and patient-reported outcomes. The optimal planning horizon, and thus the desired discount factor $\gamma^*$, can vary based on patient preference. For example, a patient might express a desire to prioritize aggressive short-term glycemic control before an important impending event, or prefer a more conservative approach at other times when they know their activity will be low. With HTI, users could then adjust the desired $\gamma^*$ based on the current clinical context. The system would then sample actions from $p(a|s, \gamma^*)$, allowing the treatment strategy to dynamically shift its focus between immediate needs and long-term objectives without retraining the entire RL policy for each desired $\gamma$.

## A.3  VARYING FIDELITY AND DIVERSITY IN GENERATIVE MODELLING

Variational Autoencoders (VAEs) (Kingma & Welling, 2014) are generative models that learn a latent representation of data. The $\beta$-VAE (Higgins et al., 2017) introduces a hyperparameter $\beta$ that modifies the VAE objective function by weighting the Kullback-Leibler (KL) divergence term, which acts as a regulariser on the latent space. The choice of $\beta$ critically influences the model's behaviour:

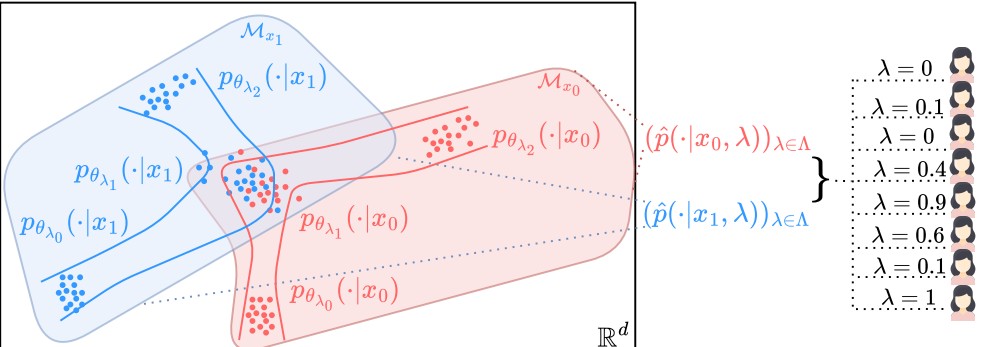

Figure 4: Example inference-time adjustment enabled by HTI. We illustrate disparate user preferences affecting desired NN behaviour (desired $\lambda$ level) for different users in this abstract example. Having a fixed number of trained NNs ($p_{\theta_{\lambda_i}}$) only allows partial exploration of the full hyperparameter trajectory, while an HTI surrogate model (($\hat{p}(\cdot|x_i, \lambda))_{\lambda \in \Lambda}$) can estimate outputs across the entire spectrum of hyperparameters (estimated conditional probability paths represented by solid blue/red lines). Crucially, hyperparameter-induced dynamics can differ amongst input conditions ($x_i$), as the true conditional distributions move along their own respective manifolds ($\mathcal{M}_{x_i}$), so an effective HTI model must learn *conditional* dynamics.

- **Low $\beta$ (e.g., $\beta < 1$):** With less pressure on the KL divergence term, the model prioritizes reconstruction accuracy. This often leads to generated samples with high *fidelity* (i.e., they closely resemble the training data and are sharp/realistic). However, the latent space might be less structured or more "entangled," potentially leading to lower *diversity* in novel generations and poorer disentanglement of underlying factors of variation.

- **High $\beta$ (e.g., $\beta > 1$):** A higher $\beta$ places more emphasis on making the learned latent distribution $q(z|x)$ close to the prior $p(z)$ (typically a standard Gaussian). This encourages a more disentangled latent space, where individual latent dimensions might correspond to distinct, interpretable factors of variation in the data (Burgess et al., 2018). While this can lead to greater *diversity* in generated samples and better generalisation for tasks like latent space interpolation, it might come at the cost of reconstruction fidelity, potentially resulting in blurrier or less detailed samples as the model sacrifices some reconstruction capacity to satisfy the stronger regularisation.

Consider a $\beta$-VAE trained to generate images. If a user needs to generate photorealistic images, a lower $\beta^*$ would be preferred to maximise the sharpness and detail, ensuring the generated image is of high perceptual quality. On the other hand, if a user is brainstorming image ideas, a higher $\beta^*$ would be beneficial, encouraging the model to generate a wider variety of images and styles, even if individual samples are slightly less photorealistic. HTI could learn a surrogate generative model $p(y_{\text{image}}|z, \beta)$. The user could then dynamically adjust $\beta^*$ based on their current task.

## A.4 FLEXIBLE HYPERPARAMETER OPTIMISATION WITH BAYESIAN OPTIMISATION

Standard Bayesian Optimization (BO) (Snoek et al., 2012; Shahriari et al., 2015) typically involves learning a probabilistic surrogate model for a specific scalar objective function $f : \Lambda \to \mathbb{R}$ (e.g., validation accuracy). This creates a rigid dependency: if the user's preference changes during deployment—for instance, shifting from maximising pure accuracy to maximising accuracy subject to a fairness constraint or an inference latency budget—the learned surrogate is no longer valid for the new objective, and the hyperparameter search process must be restarted.

HTI can decouple the surrogate model from the objective function. Because HTI learns a surrogate for $p_{\theta_\lambda}(y|x)$ rather than a scalar objective, it can be used to calculate *any* performance metric derived from the model outputs as so:

1. The HTI model is trained on a sparse set of anchor models to learn the conditional probability paths.

2. Post-training, a user can define an arbitrary objective function $\mathcal{J}(\lambda)$ based on the model outputs (e.g., Expected Calibration Error, F1-score, or a custom utility function balancing risk and reward).

3. A BO optimiser searches for the optimal $\lambda^*$ that minimises $\mathcal{J}(\lambda)$ by querying the HTI surrogate $\hat{p}(y|x, \lambda)$.

Critically, evaluating the objective $\mathcal{J}$ via the HTI surrogate is much faster than retraining the original neural network. This could allow users to explore arbitrary Pareto frontiers of competing objectives without the need for further expensive ground-truth model training, or training multiple surrogate objectives as in standard BO.

# B  SAMPLING ALGORITHM

We summarise our sampling procedure, as detailed in §4, below in Algorithm 2.

---

**Algorithm 2** Sampling from $\hat{p}(y|x, t^*)$

---

**Require:** True distributions $\{p_{t_k}(\cdot|x)\}_{t_k \in \mathcal{T}_{\text{obs}}}$, CLOT maps $\{T_{\theta_{T,k}}\}_k$, geodesic path generator $S_{\theta_S}$, target marginal $t^*$, condition $x \in \mathcal{X}$

Find $k$ such that $t_k, t_{k+1} \in \mathcal{T}_{\text{obs}}$ and $t_k < t^* < t_{k+1}$.
$y_k \sim p_{t_k}(\cdot|x)$.
$\hat{y}_{k+1} = T_{\theta_{T,k}}(y_k|x)$
Define spline geodesic path $q_\varphi(\cdot)$ with $\varphi = S_{\theta_S}(y_k, \hat{y}_{k+1}, x)$
$s^* = (t^* - t_k)/(t_{k+1} - t_k)$.                    ▷ Normalise target marginal for current interval
$\hat{y}_{t^*} = q_\varphi(s^*)$
**return** $\hat{y}_{t^*}$

---

## C EXPERIMENTAL DETAILS

We now provide detailed experimental set-ups for each of our experiments in §5. The code implementation of our method is included here: `https://github.com/harrya32/hyperparameter-trajectory-inference`.

We ran all experiments on an Azure VM A100 GPU. A single run for the semicircles experiment took between 5-10 minutes, depending on the surrogate model. To produce data for the reward-weighting experiments, it took 3.5 hours to train a PPO agent at each setting. For the cancer experiment, it took between 2-15 minutes to train the surrogate models. For the reacher experiment, it took between 1-7 minutes to train the surrogate models. For the quantile regression experiment, it took approximately 5 minutes to train the MLP quantile forecasters, and between 2-15 minutes to train the surrogate models. For the generative modelling dropout experiment, it took between 6-20 minutes to train the surrogate models. Ultimately, final experimental runs involve approximately 120 GPU hours of training time.

We base the implementation of our method off the code from Pooladian et al. (2024) (CC BY-NC 4.0 License, `https://github.com/facebookresearch/lagrangian-ot`), which we adapt for our specific setting.

### C.1 SEMICIRCLES EXPERIMENT

#### C.1.1 SEMICIRCLES DATASET

We describe here the temporal process we used to generate the conditional semicircles synthetic dataset from §5.1. The dataset comprises 2D points $(x, y)$ associated with one of four discrete conditions $c \in \{1, 2, 3, 4\}$, generated at a continuous time $t \in [0, 1]$. For each condition and time, points are generated by first sampling an angle from a Von Mises distribution (von Mises, 1918), with a time- and condition-dependent mean, and then sampling a radius from a Log-Normal distribution centred around a unit circle radius. Specifically, the generation process for a single point under condition $c$ at time $t$ is as follows:

**Global parameters:**

- $r_{\text{nom}} = 1$: Nominal radius of the semicircles.
- $\sigma_{\text{rad}} = 0.05$: Standard deviation of the logarithm of the radial component, controlling radial spread.
- $\kappa_{\text{ang}} = 5.0$: Angular concentration parameter for the Von Mises distribution.

**Generation:** For each condition $c$ and time $t$:

1. **Sample radius ($R$):** The radial component $R$ is drawn from a Log-Normal distribution, such that $\log(R)$ is normally distributed:

$$\log(R) \sim \mathcal{N}(\mu_{\text{log}}, \sigma_{\text{log}}^2)$$

where $\mu_{\text{log}} = \log(r_{\text{nom}})$ and $\sigma_{\text{log}} = \sigma_{\text{rad}}$. Thus,

$$R \sim \text{LogNormal}(\log(r_{\text{nom}}), \sigma_{\text{rad}}^2)$$

This distribution is independent of condition $c$ and time $t$.

2. **Mean angle ($\mu_{\text{ang}}(c, t)$) and semicircle centre ($x_{\text{offset}, c}$):** The mean angle $\mu_{\text{ang}}(c, t)$ and the x-coordinate of the semicircle's center $x_{\text{offset}, c}$ are determined by the condition $c$ and time $t$:

$$x_{\text{offset}, c} = \begin{cases} -1.0 & \text{if } c \in \{1, 2\} \\ 1.0 & \text{if } c \in \{3, 4\} \end{cases}$$

$$\mu_{\text{ang}}(c, t) = \begin{cases} t\pi & \text{if } c = 1 \quad \text{(top-left semicircle, } 0 \to \pi) \\ -t\pi & \text{if } c = 2 \quad \text{(bottom-left semicircle, } 0 \to -\pi) \\ (1-t)\pi & \text{if } c = 3 \quad \text{(top-right semicircle, } \pi \to 0) \\ (t-1)\pi & \text{if } c = 4 \quad \text{(bottom-right semicircle, } -\pi \to 0) \end{cases}$$

3. **Sample angle ($\Phi_c(t)$):** The angular component $\Phi_c(t)$ is drawn from a Von Mises distribution centred at the mean angle:

$$\Phi_c(t) \sim \text{VonMises}(\mu_{\text{ang}}(c, t), \kappa_{\text{ang}})$$

4. **Cartesian coordinates ($x_c(t), y_c(t)$):** The 2D coordinates are obtained by converting the sampled polar coordinates $(R, \Phi_c(t))$ to Cartesian, relative to the semicircle's center:

$$x_c(t) = x_{\text{offset},c} + R\cos(\Phi_c(t))$$
$$y_c(t) = R\sin(\Phi_c(t))$$

The full dataset at a given time $t$ consists of $N$ samples drawn from each of the four conditional distributions.

In §5.1 and §5.5 our training data consists of 100 samples for each condition at times $t \in \{0, 0.5, 1.0\}$. The geodesics plotted in Figure 1 begin at true points sampled at $t = 0$ and end at their estimated CLOT maps at $t = 1$. For the numerical results in Tables 1 and 7, we compare estimated distributions from the respective models to the true distributions at $t \in \{0.25, 0.75\}$.

### C.1.2 MODEL DETAILS

The hyperparameters for the surrogate models used in the semicircles experiments are listed in Table 8. Note that, since we have discrete conditions in this experiment, we construct separate NW density estimators for each condition, hence we set $h_x$ as N/A.

| Hyperparameter | Value |
|---|---|
| $\alpha$ | 0.05 for models with $\hat{\mathcal{U}}$, 0 otherwise |
| $h_y$ | 0.05 |
| $h_x$ | N/A |
| Epochs | 2001 |
| $G_{\theta_G}$ learning rate | $5 \times 10^{-3}$ |
| $G_{\theta_G}$ MLP hidden layer sizes | $[128, 128]$ |
| $G_{\theta_G}$ activations | ReLU |
| $G_{\theta_G}$ eigenvalue budget | 2 |
| $g_{\theta_g}, T_{\theta_T}$ MLP hidden layer sizes | $[64, 64, 64, 64]$ |
| $S_{\theta_S}$ MLP hidden layer sizes | $[1024, 1024]$ |
| $g_{\theta_g}, T_{\theta_T}, S_{\theta_S}$ learning rate | $10^{-4}$ |
| $g_{\theta_g}, T_{\theta_T}, S_{\theta_S}$ activations | ReLU |
| Spline knots | 15 |
| FiLM layer size (applied to first layer activations) | 16 |
| $c$-Transform solver | LBFGS, 10 iterations |
| Min-max optimisation | $1 \times G_{\theta_G}$ update per $10 \times g_{\theta_g}, T_{\theta_T}, S_{\theta_S}$ updates |

Table 8: Hyperparameters for semicircle experiments in §5.1.

### C.2 CANCER THERAPY EXPERIMENT

### C.2.1 ENVIRONMENT

We conduct this experiment using the 'GhaffariCancerEnv-continuous' environment from `DTR-Bench`/`DTR-Gym` (Luo et al., 2024) (`https://github.com/GilesLuo/DTRGym`, MIT license) which is based on the mathematical model for treatment of cancer with metastasis using radiotherapy and chemotherapy proposed in Ghaffari et al. (2016). The implementation deviates from Ghaffari et al. (2016) by treating the dynamics of circulating lymphocytes ($c_1$) and tumor-infiltrating cytotoxic lymphocytes ($c_2$) as constant.

The state at time $t$ is an 8-dimensional continuous vector representing key biological and treatment-related quantities:

$$S_t = [T_{p,t}, N_{p,t}, L_{p,t}, C_t, T_{s,t}, N_{s,t}, L_{s,t}, M_t]^T$$

where:

- $T_{p,t}$: Total tumour cell population at the primary site.
- $N_{p,t}$: Concentration of Natural Killer (NK) cells at the primary site (cells/L).
- $L_{p,t}$: Concentration of CD8+T cells at the primary site (cells/L).
- $C_t$: Concentration of lymphocytes in blood (cells/L).
- $T_{s,t}$: Total tumour cell population at the secondary (metastatic) site.
- $N_{s,t}$: Concentration of NK cells at the secondary site (cells/L).
- $L_{s,t}$: Concentration of CD8+T cells at the secondary site (cells/L).
- $M_t$: Concentration of chemotherapy agent in the blood (mg/L).

All state components are non-negative real values.

The action at time $t$ is a 2-dimensional continuous vector representing the treatment intensities:

$$A_t = [D_t, v_t]^T$$

where:

- $D_t$: The effect of radiotherapy applied at time $t$.
- $v_t$: The effect of chemotherapy applied at time $t$.

These actions influence the dynamics of the state variables according to the underlying mathematical ODE model.

The reward $R_t$ received after taking action $A_t$ in state $S_t$ and transitioning to state $S_{t+1}$ is designed to encourage tumor reduction while penalizing significant deviations in Natural Killer (NK) cell populations, with an additional reward or penalty in terminal states. Let $S_0 = [T_{p,0}, N_{p,0}, \dots]^T$ be the initial state of an episode. The components of the reward at each non-terminal step are:

**Tumor reduction component ($R_{\textbf{tumor}}$):** This component measures the relative reduction in total tumor cells. First, the total tumor populations at the current step $k$ (representing $S_{t+1}$) and at the initial step 0 are calculated:

$$T_{\text{tot},k} = T_{p,k} + T_{s,k} \quad \text{and} \quad T_{\text{tot},0} = T_{p,0} + T_{s,0}$$

These are then log-transformed:

$$\mathcal{T}_k = \ln(\max(e, T_{\text{tot},k})) \quad \text{and} \quad \mathcal{T}_0 = \ln(\max(e, T_{\text{tot},0}))$$

The tumor reduction reward is then:

$$R_{\text{tumor},t} = 1 - \frac{\mathcal{T}_{t+1}}{\mathcal{T}_0}$$

**NK cell population penalty ($R_{\textbf{nk}}$):** This component penalizes deviations of the total NK cell population from its initial value. The total NK cell populations are:

$$N_{\text{tot},k} = N_{p,k} + N_{s,k} \quad \text{and} \quad N_{\text{tot},0} = N_{p,0} + N_{s,0}$$

These are also log-transformed:

$$\mathcal{N}_k = \ln(\max(e, N_{\text{tot},k})) \quad \text{and} \quad \mathcal{N}_0 = \ln(\max(e, N_{\text{tot},0}))$$

The penalty is then calculated, with weighting factor $\lambda_{\text{nk}}$:

$$R_{\text{nk},t} = -\lambda_{\text{nk}} \left| \frac{\mathcal{N}_{t+1}}{\mathcal{N}_0} - 1 \right|$$

Finally, a **termination reward ($R_{\textbf{term}}$)** is added if the episode ends:

$$R_{\text{term}} = \begin{cases} 100 & \text{if positive termination (no more tumour)} \\ -100 & \text{if negative termination (max tumour size)} \\ 0 & \text{if non-terminal step} \end{cases}$$

The total reward at step $t$ is:

$$R_t = R_{\text{step},t} + R_{\text{term}} = \left(1 - \frac{\mathcal{T}_{t+1}}{\mathcal{T}_0}\right) - \lambda_{\text{nk}} \left| \frac{\mathcal{N}_{t+1}}{\mathcal{N}_0} - 1 \right| + R_{\text{term}}$$

### C.2.2 Non-linear Reward Variant:

For the non-linear reward scalarization experiment (§5.2.3), denoted as `Cancer_nonlin`, we modify the reward function to incorporate a hinge mechanism on the NK penalty term, employing the non-linear reward scalarization discussed in eq. (6) in (Rădulescu et al., 2020). In this setting, the weighted NK cell penalty is only active if the relative deviation exceeds a threshold of 0.01. The modified penalty term $R_{\text{nk},t}$ is defined as:

$$R_{\text{nk},t} = \begin{cases} -\lambda_{\text{nk}} \left| \frac{\mathcal{N}_{t+1}}{\mathcal{N}_0} - 1 \right| & \text{if } \left| \frac{\mathcal{N}_{t+1}}{\mathcal{N}_0} - 1 \right| > 0.01 \\ 0 & \text{otherwise} \end{cases}$$

All other components of the reward function remain unchanged.

### C.2.3 Policies

We train PPO agents (Schulman et al., 2017) for the true distributions $p_{\theta_\lambda}(a|s)$ at various $\lambda_{\text{nk}}$ settings using the implementation in `Stable Baselines3` (Raffin et al., 2021) (MIT license, https://github.com/DLR-RM/stable-baselines3), with all other hyperparameters left at default, using the `MLPPolicy` architecture. For each agent, we train for $500,000$ timesteps.

Once trained, we use samples from the models with $\lambda_{\text{nk}} \in \{0, 5, 10\}$ as the training dataset for each surrogate model. Specifically, we run the agent with $\lambda_{\text{nk}} = 10$ for 100 steps in the environment, collecting 10 actions from each (stochastic) policy per observation. We evaluate each surrogate model at $\lambda_{\text{nk}} \in \{1, 2, 3, 4, 6, 7, 8, 9\}$.

### C.2.4 Model details

The hyperparameters for our surrogate models for the adaptive reward-weighting experiment are listed in Table 9.

| Hyperparameter | Value |
|---|---|
| $\alpha$ | 0.01 for models with $\hat{\mathcal{U}}$, 0 otherwise |
| $h_y$ | 1.0 |
| $h_x$ | 1.0 |
| Epochs | 2001 |
| $G_{\theta_G}$ learning rate | $5 \times 10^{-3}$ |
| $G_{\theta_G}$ MLP hidden layer sizes | $[128, 128]$ |
| $G_{\theta_G}$ activations | ReLU |
| $G_{\theta_G}$ eigenvalue budget | 2 |
| $g_{\theta_g}, T_{\theta_T}$ MLP hidden layer sizes | $[64, 64, 64, 64]$ |
| $S_{\theta_S}$ MLP hidden layer sizes | $[1024, 1024]$ |
| $g_{\theta_g}, T_{\theta_T}, S_{\theta_S}$ learning rate | $10^{-4}$ |
| $g_{\theta_g}, T_{\theta_T}, S_{\theta_S}$ activations | ReLU |
| Spline knots | 15 |
| FiLM layer size (applied to first layer activations) | 16 |
| $c$-Transform solver | LBFGS, 3 iterations |
| Min-max optimisation | $1 \times G_{\theta_G}$ update per $10 \times g_{\theta_g}, T_{\theta_T}, S_{\theta_S}$ updates |

Table 9: Hyperparameters for our surrogate models in the cancer therapy experiment in §5.2.1 and §5.2.3.

For the direct surrogate model, we train a four-layer MLP using supervised learning, with inputs of the base action, condition, and target hyperparameter, and output of the target action at the relevant hyperparameter setting. We list the direct surrogate hyperparameters in Table 10.

For the CFM surrogate model (Lipman et al., 2023), we train two flow matching models, to model the vector fields between the distributions at $\lambda_{\text{nk}} = 0$ and $\lambda_{\text{nk}} = 5$, and between $\lambda_{\text{nk}} = 5$ and $\lambda_{\text{nk}} = 10$ respectively. We base our implementation on the open source code from Lipman et al. (2024), found here: https://github.com/facebookresearch/flow_matching (CC BY-NC

| Hyperparameter | Value |
|---|---|
| Epochs | 10000 |
| Early stopping patience | 100 |
| Validation set | 10% |
| Batch size | 256 |
| Learning rate | $10^{-3}$ |
| Hidden layer sizes | $[64, 64, 64, 64]$ |
| Activation function | Swish |
| FiLM layer size (applied to first layer activations) | 16 |

Table 10: Hyperparameters for the direct surrogate model in the cancer therapy experiment in §5.2.1.

4.0 License). We extend this implementation to incorporate external conditions via a FiLM layer. The hyperparameters for both of the CFM models in this surrogate model are listed in Table 11.

| Hyperparameter | Value |
|---|---|
| Epochs | 10000 |
| Early stopping patience | 100 |
| Validation set | 10% |
| Batch size | 1000 |
| Learning rate | $10^{-3}$ |
| Hidden layer sizes | $[64, 64, 64, 64]$ |
| Activation function | Swish |
| FiLM layer size (applied to first layer activations) | 16 |

Table 11: Hyperparameters for the CFM surrogate model in the cancer treatment experiment in §5.2.1.

For the MFM surrogate model, we base our implementation on the open source code from Kapusniak et al. (2024), found here: `https://github.com/kkapusniak/metric-flow-matching` (MIT License). This method is similar to flow matching, however it aims to learn a vector field that leads to interpolants staying on the data manifold defined by the observed data. It does so by learning a NN-based correction to the straight line interpolants used in CFM training, that is designed to minimise the transport cost associated with a data-dependent Riemannian metric. These corrected interpolants are then used to train a neural vector field.

For the data-depended metric, we use their LAND formulation, which sets a diagonal metric $G_{\text{LAND}}(x) = \text{diag}(h(x) + \varepsilon I)^{-1}$, where

$$h_\alpha(x) = \sum_{i=1}^{N} (x_i^\alpha - x^\alpha)^2 \exp(-\frac{||x - x_i||^2}{2\sigma^2}), \tag{16}$$

with kernel size $\sigma$.

We extend the original implementation to incorporate external conditions via a FiLM layer in both the NN-based interpolant correction, and the neural vector field. The hyperparameters we use for the MFM method are listed in Table 12.

For the NLOT surrogate model, this is equivalent to our method, but setting $\hat{\mathcal{U}} = 0$ and using the fixed eigenvalue metric from Pooladian et al. (2024). We use the same hyperparameters as our surrogate models for this method.

| Hyperparameter | Value |
|---|---|
| Epochs | 2000 |
| Early stopping patience | 100 |
| Validation set | 10% |
| Batch size | 128 |
| Interpolant learning rate | $10^{-4}$ |
| Vector field learning rate | $10^{-3}$ |
| Interpolant hidden layer sizes | $[64, 64, 64]$ |
| Vector field hidden layer sizes | $[64, 64, 64]$ |
| Activation function | SELU |
| FiLM layer size (applied to first layer activations) | 16 |
| $\varepsilon$ | 0.001 |
| $\alpha$ | 1 |
| $\sigma$ | 3 |

Table 12: Hyperparameters for the MFM surrogate model in the cancer treatment experiment in §5.2.1.

## C.3 REACHER

### C.3.1 ENVIRONMENT

We conduct this experiment using the `Reacher-v2` environment from OpenAI Gym (`https://github.com/openai/gym`, MIT License). This environment consists of a two-jointed robotic arm where the goal is to move the arm's end-effector to a randomly generated target location.

The state at time $t$ is an 11-dimensional continuous vector representing the angles and velocities of the arm's joints, as well as the location of the target and the vector from the fingertip to the target:

$$S_t = [\cos(\theta_1), \cos(\theta_2), \sin(\theta_1), \sin(\theta_2), x_{\text{target}}, y_{\text{target}}, \dot{\theta}_1, \dot{\theta}_2, x_{\text{fingertip}} - x_{\text{target}}, y_{\text{fingertip}} - y_{\text{target}}, z_{\text{fingertip}} - z_{\text{target}}]^T$$

where:

- $\cos(\theta_1), \cos(\theta_2)$: Cosine of the angles of the two joints.
- $\sin(\theta_1), \sin(\theta_2)$: Sine of the angles of the two joints.
- $x_{\text{target}}, y_{\text{target}}$: The x and y coordinates of the target location.
- $\dot{\theta}_1, \dot{\theta}_2$: The angular velocities of the two joints.
- $x_{\text{fingertip}} - x_{\text{target}}, y_{\text{fingertip}} - y_{\text{target}}, z_{\text{fingertip}} - z_{\text{target}}$: The vector from the fingertip to the target.

The action at time $t$ is a 2-dimensional continuous vector representing the torque applied to the two joints:

$$A_t = [\tau_1, \tau_2]^T$$

and each $\tau_i \in [-1, 1]$.

The reward $R_t$ received at each step is the sum of a distance-to-target reward and a control cost penalty:

$$R_t = -\|\vec{p}_{\text{fingertip}, t+1} - \vec{p}_{\text{target}}\|_2 - \lambda_{\text{control}}\|\vec{a}_t\|_2^2$$

where the first term is the negative Euclidean distance between the fingertip and the target, and the second is the negative squared Euclidean norm of the action vector, which penalises large torques. We introduce the weighting hyperparameter, $\lambda_{\text{control}}$, that controls the strength of the control penalty in the reward.

### C.3.2 POLICIES

We train PPO agents (Schulman et al., 2017) at the setting $\lambda_{\text{control}} \in \{1, 2, 3, 4, 5\}$ using the implementation in `Stable Baselines3` (Raffin et al., 2021) (MIT license, `https://github.`

`com/DLR-RM/stable-baselines3`). We use the `MLPPolicy` architecture with default hyperparameters. Each agent is trained for $1,000,000$ total timesteps.

Once trained, we use samples from the models with $\lambda_{\text{control}} \in \{1, 5\}$ as the training dataset for our surrogate model $\hat{p}(a|s, \lambda)$. Specifically, we run the agent with $\lambda_{\text{control}} = 1$ for 1000 steps in the environment, collecting actions from each policy per observation. We evaluate each surrogate model at $\lambda_{\text{control}} \in \{2, 3, 4\}$.

### C.3.3 MODEL DETAILS

The hyperparameters for our surrogate models for the Reacher experiment are listed in Table 13.

| Hyperparameter | Value |
|---|---|
| $\alpha$ | 0.001 for models with $\hat{\mathcal{U}}$, 0 otherwise |
| $h_y$ | 2.0 |
| $h_x$ | 1.0 |
| Epochs | 2001 |
| $G_{\theta_G}$ learning rate | $5 \times 10^{-3}$ |
| $G_{\theta_G}$ MLP hidden layer sizes | $[128, 128]$ |
| $G_{\theta_G}$ activations | ReLU |
| $G_{\theta_G}$ eigenvalue budget | 2 |
| $g_{\theta_g}, T_{\theta_T}$ MLP hidden layer sizes | $[64, 64, 64, 64]$ |
| $S_{\theta_S}$ MLP hidden layer sizes | $[1024, 1024]$ |
| $g_{\theta_g}, T_{\theta_T}, S_{\theta_S}$ learning rate | $10^{-4}$ |
| $g_{\theta_g}, T_{\theta_T}, S_{\theta_S}$ activations | ReLU |
| Spline knots | 15 |
| FiLM layer size (applied to first layer activations) | 16 |
| $c$-Transform solver | LBFGS, 3 iterations |
| Min-max optimisation | $1 \times G_{\theta_G}$ update per $10 \times g_{\theta_g}, T_{\theta_T}, S_{\theta_S}$ updates |

Table 13: Hyperparameters for reacher experiment in §5.2.2.

For the direct surrogate model, we train a four-layer MLP in the same fashion as the cancer therapy experiment, with the same hyperparameters (Table 10).

For the CFM surrogate model, we train one flow matching model, between the distributions at $\lambda_{\text{control}} = 1$ and $\lambda_{\text{control}} = 5$ with the same hyperparameters as in the cancer experiment (Table 11).

For the MFM surrogate model, we train with the same hyperparameters as in the cancer experiment (Table 12), but with $\sigma = 1$.

### C.4 QUANTILE REGRESSION

### C.4.1 DATA

We use the ETTm2 dataset from the Electricity Transformer Temperature (ETT) collection (Zhou et al., 2021) (`https://github.com/zhouhaoyi/ETDataset`, CC BY-ND 4.0 License), which contains data on electricity load and oil temperature. We formulate a forecasting task for oil temperature, with an input horizon of 12 steps to predict an output horizon of 3 steps. The dataset is partitioned chronologically, with the first $70\%$ used for training the ground-truth models and the subsequent $15\%$ for validation. From the remaining data, the next 1200 samples form the training set for the HTI surrogates, and the final 180 samples are used as the HTI testing set to evaluate surrogate model performance.

### C.4.2 GROUND-TRUTH FORECASTERS

The ground-truth forecasters are three-layer MLPs with hidden dimensions of $[256, 128, 128]$. We train a separate model for each target quantile $\tau \in \{0.01, 0.1, 0.25, 0.5, 0.75, 0.9, 0.99\}$. Training is performed for up to 2000 epochs using the pinball loss function, with a learning rate of $10^{-3}$ and a batch size of 32. We employ early stopping with a patience of 10 epochs.

The pinball loss, $L_\tau(y, \hat{y})$, for a true value $y$ and a quantile forecast $\hat{y}$ at quantile level $\tau$ is defined as:

$$L_\tau(y, \hat{y}) = \begin{cases} \tau(y - \hat{y}) & \text{if } y \geq \hat{y} \\ (1 - \tau)(\hat{y} - y) & \text{if } y < \hat{y} \end{cases}$$

This loss function penalizes under-prediction and over-prediction asymmetrically, which encourages the model to learn the specified quantile.

To create the HTI training dataset, we use the ground-truth forecasters trained for $\tau = 0.01$ and $\tau = 0.99$ to generate forecasts on the 1200 inputs of the HTI training set. For evaluation, the forecasts from the remaining ground-truth models (for $\tau \in \{0.1, 0.25, 0.5, 0.75, 0.9\}$) on the 180 HTI test inputs serve as the ground-truth quantiles.

### C.4.3 MODEL DETAILS

The hyperparameters for our surrogate models for the quantile regression experiment are listed in Table 14.

| Hyperparameter | Value |
|---|---|
| $\alpha$ | 0.01 for models with $\hat{\mathcal{U}}$, 0 otherwise |
| $h_y$ | 1.0 |
| $h_x$ | 1.0 |
| Epochs | 1001 |
| $G_{\theta_G}$ learning rate | $5 \times 10^{-3}$ |
| $G_{\theta_G}$ MLP hidden layer sizes | $[128, 128]$ |
| $G_{\theta_G}$ activations | ReLU |
| $G_{\theta_G}$ eigenvalue budget | 3 |
| $g_{\theta_g}, T_{\theta_T}$ MLP hidden layer sizes | $[64, 64, 64, 64, 64, 64, 64, 64]$ |
| $S_{\theta_S}$ MLP hidden layer sizes | $[1024, 1024]$ |
| $g_{\theta_g}, T_{\theta_T}, S_{\theta_S}$ learning rate | $10^{-4}$ |
| $g_{\theta_g}, T_{\theta_T}, S_{\theta_S}$ activations | ReLU |
| Spline knots | 15 |
| FiLM layer size (applied to first layer activations) | 16 |
| $c$-Transform solver | LBFGS, 10 iterations |
| Min-max optimisation | $1 \times G_{\theta_G}$ update per $10 \times g_{\theta_g}, T_{\theta_T}, S_{\theta_S}$ updates |

Table 14: Hyperparameters for ETT experiment in §5.3.

For the direct surrogate model, we train an eight-layer MLP with a hidden dimension of 64, to match the increase in the number of layers for the Kantorovich potential and CLOT map MLPs in our surrogate models. The other hyperparameters are the same as in the cancer experiment (Table 10).

For the CFM surrogate model, we also use an eight-layer MLP with a hidden dimension of 64 for the flow matching model. The other hyperparameters are the same as in the cancer experiment (Table 11).

For the MFM surrogate model, we train with the same hyperparameters as in the cancer experiment (Table 12), but with $\sigma = 0.01$.

### C.5 DROPOUT EXPERIMENT

### C.5.1 DATA

The two-moons dataset was generated with `sklearn.datasets.make_moons` using $n = 2000$ points and noise of $0.05$. The 2D co-ordinates were standardised to zero mean and unit variance prior to any modelling; the binary class label $x \in 0, 1$ was used as the conditioning variable. For the HTI training set we trained ground-truth diffusion models at anchor dropout levels $p \in \{0, 0.5, 0.99\}$ and drew 1000 samples from each converged model (balanced across classes).

### C.5.2 DIFFUSION MODEL

Ground-truth models were implemented as DDPMs (Ho et al., 2020) with a conditional MLP score network. The network receives the noisy coordinates $y \in \mathbb{R}^2$, a normalised timestep scalar $t$ and the class condition $x$, and predicts the noise $\varepsilon \in \mathbb{R}^2$. The MLP has two hidden layers of size 256, ReLU activations and dropout applied after each hidden layer. The forward schedule uses $T = 100$ timesteps with $\beta$ linearly spaced from $1\mathrm{e}{-4}$ to $0.02$. Models were trained with Adam (learning rate $1\mathrm{e}{-3}$), batch size 128, and for 250 epochs. Sampling was performed via the standard ancestral reverse diffusion loop using the trained network and the same noise schedule.

### C.5.3 MODEL DETAILS

Surrogate HTI models were trained to interpolate between the three anchor diffusion marginals and evaluated at held-out dropout settings $p \in \{0.1, 0.2, 0.3, 0.4, 0.6, 0.7, 0.8, 0.9\}$. Evaluation compares the Wasserstein distance between surrogate pushforward samples and samples from ground-truth diffusion models trained at the target $p$ values. Hyperparameters used to train our surrogates are listed in Table 15.

| Hyperparameter | Value |
|---|---|
| $\alpha$ | 0.01 for models with $\hat{\mathcal{U}}$, 0 otherwise |
| $h_y$ | 0.2 |
| $h_x$ | 1.0 |
| Epochs | 2001 |
| $G_{\theta_G}$ learning rate | $5 \times 10^{-3}$ |
| $G_{\theta_G}$ MLP hidden layer sizes | $[128, 128]$ |
| $G_{\theta_G}$ activations | ReLU |
| $G_{\theta_G}$ eigenvalue budget | 3 |
| $g_{\theta_g}, T_{\theta_T}$ MLP hidden layer sizes | $[64, 64, 64, 64]$ |
| $S_{\theta_S}$ MLP hidden layer sizes | $[1024, 1024]$ |
| $g_{\theta_g}, T_{\theta_T}, S_{\theta_S}$ learning rate | $10^{-4}$ |
| $g_{\theta_g}, T_{\theta_T}, S_{\theta_S}$ activations | ReLU |
| Spline knots | 15 |
| FiLM layer size (applied to first layer activations) | 16 |
| $c$-Transform solver | LBFGS, 10 iterations |
| Min-max optimisation | $1 \times G_{\theta_G}$ update per $10 \times g_{\theta_g}, T_{\theta_T}, S_{\theta_S}$ updates |

Table 15: Hyperparameters for dropout experiment in §5.4.

For the direct surrogate model, we train a four-layer MLP in the same fashion as the cancer therapy experiment, with the same hyperparameters (Table 10).

For the CFM surrogate model with the same hyperparameters as in the cancer experiment (Table 11).

For the MFM surrogate model, we train with the same hyperparameters as in the cancer experiment (Table 12), but with $\sigma = 0.5$.

## D    EXTENDING TO MULTIPLE HYPERPARAMETERS

A limitation of our current approach is its design is only immediately appropriate for a single, continuous hyperparameter. We see extensions to multiple and discrete hyperparameter settings as a key direction for future research. Extending our current HTI method to this setting is non-trivial.

One simple extension that would allow for interpolation between multiple hyperparameters with our current method involves extablishing a mapping from the multi-dimensional hyperparameter space to a single 'time' space, allowing our interpolation scheme that works on a single dimensional 'time' variable to apply. We have considered two representative strategies for creating such a mapping—a data-driven Principal Curve and a geometric space-filling Hilbert Curve—but there are outstanding limitations to both potential approaches.

- Principal Curves: A principal curve, a non-linear generalisation of PCA, is the smooth curve that captures the most variance a dataset. If we have multiple observed multi-dimensional hyperparameters, we could find the principle curve through them, which could serve as our 1D 'time' axis. The primary limitation of this approach is that it only allows for interpolation to hyperparameter settings defined along this learned curve. To approximate an arbitrary setting that is not on the curve, one would first have to project it onto the curve.

- Hilbert Curves: Conversely, a space-filling Hilbert Curve is a pre-defined geometric construction whose single, continuous line is guaranteed to pass through every point in a multi-dimensional space, ensuring full coverage. While this could guarantee coverage, its critical flaw is that it breaks locality. Our method is grounded in Optimal Transport and least-action principles, which assume that small changes in our "time" variable should lead to small changes in the output distribution. A Hilbert curve would not necessarily respect this intuition, potentially mapping two very different distributions to be 'temporal neighbours'.

# E  SPARSITY INVESTIGATION

To investigate sensitivity of different surrogates to data sparsity, we evaluate HTI performance with various number of anchor distributions in the `Cancer` and `Reacher` environments. We range from the sparse settings from §5.2.1 and §5.2.2 to a dense setting, where training data is available at every evaluation setting. Figure 5 shows that, in both environments, the performance gap between methods is negligible in the dense regime, where interpolation is trivial, and this widens as sparsity increases. Our method degrades the least as interpolation becomes more difficult, confirming the effect of our inductive biases and their importance in sparse settings.

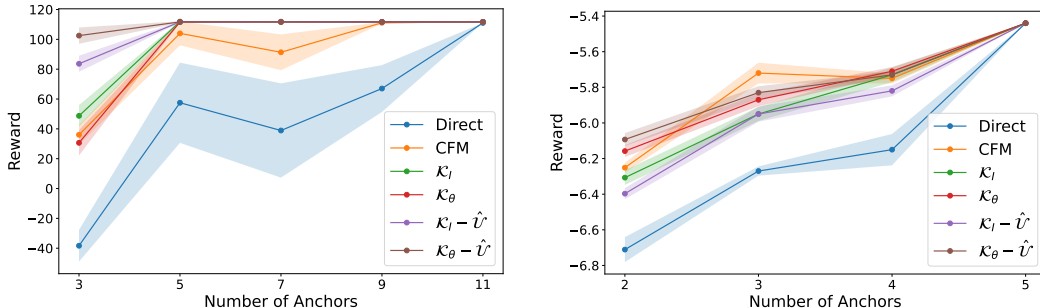

Figure 5: Surrogate model reward in `Cancer` (left) and `Reacher` (right).

## F    LLM USAGE

In this work, we used LLMs to assist with the writing of this manuscript. This primarily involved consulting LLMs to refine drafts, improving the coherence and clarity of our work, and simplifying the writing process.

We also used LLMs during the experimental process, to help write code.

