# OpenReview forum: "Hyperparameter Trajectory Inference with Conditional Lagrangian Optimal Transport"
_ICLR.cc/2026/Conference — ICLR 2026 Oral_

### Official Review · Reviewer_Ei6o · 2025-10-26

**Soundness:** 2
**Presentation:** 1
**Contribution:** 2
**Rating:** 2
**Confidence:** 3

**Summary:**

This paper introduces a technique for performing hyper-parameter trajectory inference (HTI), a problem where one introduces a model $\hat{p}(y|x,\lambda) \approx p\_{\theta\_{\lambda}}(y|x)$, which eases the adjustment of $\lambda$ at inference time. To that end, the authors propose using Lagrangian Optimal Transport (Villani, 2009; Pooladian et al., 2024) to build their model $\hat{p}$.

**Strengths:**

There are 2 main strong points with this submission.

First, the authors do a nice link in their experiments section, with their motivations in the introduction. The reasoning for why one needs hyper-parameter inference are also clear and well motivated.

Second, there is a good variety of important tasks for which the proposed method applies. In all tasks the authors show an improvement over other methods, sometimes with a significant margin.

**Weaknesses:**

Overall, I think this paper has major organization issues:

1) Section 4 is way too short to merit being a section on its own (2 paragraphs), and I think the authors define the problem they are treating way too late in the paper. In my view, this should be done as early as possible (e.g., in the beginning preliminaries section).

2) With `1)` in mind, the preliminaries section seem disconnected with the problem statement. It could be nice to highlight how conditional OT/lagrangian OT relate to the HTI task.

3) The paper seems to have many interesctions with the work of (Pooladian et al., 2024). For instance, they use the same amortization procedure and metric parametrisation. With that in mind, it would be nice to have a comparison with their method, highlighting the improvements over previous art.

4) About section 5 in general, the fact that the main algorithm is only shown in the appendix harms readability. Furthermore, the authors don't present the final objective function/learning problem being optimized.

**Questions:**

The hyper-parameter trajectory notion seems a bit contradictory to me. For instance, usually hyper-parameters are fixed during training, so they don't really evolve. From what I got from the text, the authors are trying to model the mapping $\lambda \mapsto p\_{\theta\_{\lambda}}$ (see the intro) but in my view this lacks a temporal structure, i.e., $\lambda(t), t \in [0 , 1]$ to warrant the nomenclature "trajectory inference". Could the authors elaborate on this idea?

---

> ### Author Response · Authors · 2025-11-23
>
> Thank you for your thoughtful comments and suggestions, which have helped us significantly improve our manuscript. Please see the revised PDF, with changes in $\textcolor{blue}{\text{blue}}$. We also directly respond to your specific points below.
>
> ---
> **(A) Organisation changes**
>
> We agree with your assessment that the problem definition appeared too late in the original manuscript and would be better integrated with the mathematical preliminaries. We have revised the manuscript to improve flow and clarity:
>
> 1. We have moved the introduction of **Hyperparameter Trajectory Inference (HTI)** into the beginning of **Section 2**.
> 2. We have restructured the preliminaries to better link the mathematical framework of Conditional and Lagrangian OT to the specific requirements and use case of HTI, better motivating the mathematical background by the problem statement.
> 3. We have made Algorithm 1, which describes our training procedure, more comprehensive, and moved it into the main body in **Section 4**. We would like to clarify that our ultimate objective is described in Equation 12, which involves a min-max optimisation. The introduction of the amortised CLOT maps, and geodesic spline predictors, allow us to efficiently conduct this optimisation, as now described in Algorithm 1.
>
> **Update:** These **structural changes** are reflected throughout the revised **Sections 2 and 4**.
>
> ---
> **(B) Comparison with Pooladian et al., 2024**
>
> We respectfully clarify that our metric parametrisation is **not** the same as that of Pooladian et al. (2024), as indeed our novel parametrisation is one of our main contributions. The parametrisation in Pooladian et al. (2024) is restricted to 2D ambient spaces, with fixed anisotropy at any point, since they fix the eigenvalues of their neural metric to be 1 and 0.1. In contrast, our approach allows for flexible, learned anisotropy that extends to higher-dimensional spaces. We achieve this by predicting the eigenvalues of the metric with a neural network, while ensuring that they sum to a positive 'eigenvalue budget'.
>
> While our original manuscript had already included an ablation of this specific component (now in Section 5.4), we have now added a further comprehensive comparison against the full NLOT method from Pooladian et al. (2024). Since NLOT is designed for unconditional TI, we adapted it for a fair comparison by equipping their networks with FiLM conditioning layers (identical to our method's approach to conditioning). The results on the 2D experiments are shown below:
>
> | Method | Semicircle NLL ($\downarrow$) | Semicircle CD ($\downarrow$) | Cancer Reward ($\uparrow$) | Reacher Reward ($\uparrow$) |
> | :--- | :---: | :---: | :---: | :---: |
> | NLOT (Pooladian et al. (2024)) | $13.293$ $(1.98)$ | $0.159$ $(0.008)$ | $9.26$ $(10.55)$ | $-6.173$ $(0.038)$ |
> | Ours | $-0.662$ $(0.046)$ | $0.016$ $(0.001)$ | $102.49$ $(5.46)$ | $-6.093$ $(0.036)$ |
>
> Our method significantly outperforms NLOT in each experiment. This performance is due to the impact of two contributions: 1) the incorporation of our density-based potential energy term $\hat{\mathcal{U}}$ into the Lagrangian, and 2) our more expressive parametrisation of the learned metric $G_{\theta_G}$. Furthermore, our method is capable of extending beyond 2D ambient dimensions, which NLOT inherently cannot.
>
> **Update:** We have **added this explicit comparison** and discussion to **Appendix E**.
>
> ---
> **(C) Link to trajectory inference**
>
> To clarify the "trajectory" terminology, we note that you are correct that hyperparameters are usually fixed during training, and this is the setting we operate in. We use the term "trajectory" to describe the **continuum of optimal NN output distributions** defined by a single continuous hyperparameter. We do not consider this hyperparameter as changing during the training of a single model. Rather, we view the continuous hyperparameter $\lambda$ as acting as "time" in the trajectory analogy. As $\lambda$ varies, the resulting converged output distribution $p_{\theta_\lambda}(y|x)$ moves along a manifold; recovering this path is the goal of our inference.
>
> **Update:** We have **refined Section 2 to better define this analogy** and clarify that we are modelling the outputs of converged NN distributions.
>
> ---
>
> Thank you for your time and effort. We hope that the reorganised manuscript and the new experimental comparisons address your concerns. We are more than happy to engage with any further comments you may have.

---

> > ### Comment · Reviewer_Ei6o · 2025-11-24
> > **Official Response to Authors' Rebuttal**
> >
> > Dear authors,
> >
> > Thank you for your work and effort during the rebuttal.
> >
> > I think the paper is much stronger with the modifications, and the comparison (and improvements) with respect NLOT provides a clear improvement over previous SOTA. I think the current version is good and ready for publication. Therefore, I am changing my score from __2. Reject, not good enough__ to __8, Accept (poster)__. Congratulations on your work.
> >
> > __Note.__ The provided results in the table in the rebuttal were not incorporated to the main paper for some reason. The authors should include them in the final version.

---

> > > ### Author Response · Authors · 2025-11-24
> > >
> > > Thank you very much for your continued engagement with our work! Your comments have certainly helped strengthen the manuscript, and we are glad that we have addressed all of your concerns. We are very grateful for the score increase! We will indeed incorporate these NLOT results from the Appendix into the main paper, as new rows in each of the main experimental tables.

---

### Official Review · Reviewer_rFXY · 2025-10-31

**Soundness:** 4
**Presentation:** 3
**Contribution:** 4
**Rating:** 6
**Confidence:** 3

**Summary:**

This paper studies the problem of hyperparameter optimization for training neural networks. To do this, the authors develop a conditional Lagrangian optimal transport framework that allows them to learn a Lagrangian for problems of interest. The authors demonstrate that the learned Lagrangian can accurately model hyperparameter landscapes across examples, demonstrating the utility of their method.

**Strengths:**

- The paper does a good job of setting up its methodology, with clear explanations of each piece.
- The problem formulation developed here seems new and is well motivated.
- The use of learning a conditional Lagrangian Optimal Transport model is novel and elegant.
- The authors give a range of experiments that demonstrate the ability of this method to learn hyperparameter trajectories.

**Weaknesses:**

- It is hard to see what the real use of this method will be, and the message of this gets somewhat lost in the experiments.  While this is a cool idea, what are the applications in which this will be most relevant, especially when one doesn't know a priori how chaotic the trajectories will be?
- There are no baselines that this work is able to compare against.
- While the results indicate that the method can do a good job of modeling trajectories with proper tuning, I am missing what insights we gain from these trajectories. Rather than just chasing performance, can the authors comment on interesting findings in what is actually learned?

**Questions:**

- Rather than just empirical evidence across examples, can the authors give a theoretically motivated way to choose which method to use? In other words, how do we guide our choice of conditional Lagrangian? Is there an automatic way to do this?
- Does the method do a good job of extrapolating outside of the range of learned hyperparameters? Having a clear discussion of interpolation versus extrapolation in trajectory inference like this is useful.
- I cannot get a sense of how scalable this method is. How expensive is this to run in really large scale ML settings?
- Can the authors combine their work with Bayesian optimization formulations to yield a way to choose optimal hyperparameters rather than just modeling across a range of hyperparameters?

---

> ### Author Response · Authors · 2025-11-23
> **Part 1/2**
>
> Thank you for your thoughtful comments and suggestions, which have helped us significantly improve our manuscript. Please see the revised PDF, with changes in $\textcolor{blue}{\text{blue}}$. We also directly respond to your specific points below.
>
> ---
> **(A) Utility of HTI**
>
> We emphasise that the primary use case of HTI, that we use to motivate our work, is **efficient inference-time adaptation**.
>
> To better quantify the time benefits that HTI can confer, let us consider Figure 2, which displays the behaviour of different policies (real and surrogate) for our `Cancer` example. This plot displays the behaviour of networks at 11 hyperparameter settings. Obtaining the ground-truth results requires training 11 separate PPO policies, taking approximately **38 GPU hours**, while the surrogate result requires training only three PPO policies and an HTI surrogate, taking approximately **11 GPU hours**. This is a substantial gain in efficiency (approx. $3.5\times$ speedup for this instance). In domains like healthcare, where optimal behaviours are highly context-specific, HTI can enable rapid personalisation that would otherwise be computationally infeasible.
>
> **Update:** We have included this **explicit time-saving breakdown** in **Section 6**.
>
> ---
> **(B) Baselines**
>
> We respectfully clarify that while HTI is a novel problem formulation, we **do** compare against strong, applicable baselines in **Section 5**:
> 1.  **Conditional Flow Matching (CFM):** A current state-of-the-art method for generative modeling and trajectory inference.
> 2.  **Direct Supervised Regression:** A standard MLP baseline.
> 3.  **Ablations:** We compare against Euclidean OT ($K_I$) and metric-only OT ($\mathcal{K}_\theta$), and potential-only OT ($\mathcal{K}_I - \hat{\mathcal{U}}$) to strictly isolate the benefits of our proposed Lagrangian components.
>
> Additionally, we have now added a comparison to the exact **Neural Lagrangian OT (NLOT)** method from Pooladian et al. (2024) in **Appendix E**, which our method also significantly outperforms.
>
> **Update:** We have **added further results** against the NLOT method from Pooladian et al. (2024) **Appendix E**.
>
> ---
> **(C) Trajectory insights**
>
> To investigate how HTI can be used to gain insights from the learned trajectories, beyond raw performance, we conducted a **Sparsity Investigation** (**Section 5.3**).
>
> By evaluating how well methods reconstruct trajectories as we add/remove anchor points, we gain insight into the *complexity* of the hyperparameter dynamics. If a trajectory can be well-reconstructed in sparse settings, it is likely easier to model/displays less complex dynamics that one which can only be reconstructed from dense observations. To analyse this we conducted a new ablation in the `Cancer` and `Reacher` environments, training HTI surrogates with data from varying numbers of anchor policies. From Figure 3, we can see that:
> *   In the `Reacher` experiment, performance degrades consistently as anchors are removed, suggesting the hyperparameter $\lambda_{control}$ has a large effect on the converged network, in a way that requires frequent sampling to capture well.
> *   In the `Cancer` experiment, however, most methods perform well until the very sparsest setting, where performance can drop sharply. This suggests the $\lambda_{NK}$-induced dynamics are smoother, and easier to reconstruct from sparse samples.
>
> **Update:** We have added this sparsity analysis to **Section 5.3**.
>
> ---
> **(D) How to choose Lagrangian settings**
>
> Because the dynamics of neural network hyperparameters are generally unknown a priori, there is no closed-form theoretical guarantee for the "correct" Lagrangian. However, we can provide some practical recommendations:
>
> 1.  We advise using the most expressive setting ($\mathcal{K}_\theta - \hat{\mathcal{U}}$) as default, as it consistently performed best across all our experiments in Section 5.
> 2.  For the potential weight $\alpha$, we recommend a standard cross-validation approach if possible: withhold a subset of anchor distributions (or a middle anchor) during validation, and select the $\alpha$ that maximises performance at this unseen setting. In all our experiments, $\alpha$ was kept generally small ($0.01 - 0.05$).
>
> ---
> **(E) Extrapolation beyond observed hyperparameters**
>
>
> As our method is grounded in Optimal Transport, HTI is fundamentally an **interpolation** framework. It transports mass from a source distribution to a target distribution. It is therefore not designed to, or able to, extrapolate $\lambda$ beyond the range of observed anchors.

---

> ### Author Response · Authors · 2025-11-23
> **Part 2/2**
>
> **(F) Scalability**
>
> Our metric parameterization is the main consideration when scaling this method to larger models with bigger ambient dimensions. The output dimension of $G_{\theta_{G}}$ scales quadratically with the ambient dimension $D_y$ (due to the number of Givens rotations it must parametrise). We successfully applied the method to quantile regression with 12 dimensional inputs, and a 3 dimensional ambient space, and in this setting training took only ~15 minutes. Training cost also scales with the number of anchor points, scaling linearly with the number of interval pairs, since new potential and transport map networks are needed for each adjacent pair. However, HTI is specifically designed for the **sparse** setting, where the number of anchor points is low. HTI provides maximum value specifically when training ground-truth models is expensive, keeping the number of anchor distributions small.
>
>
> ---
> **(G) Bayesian Optimisation**
>
> Thanks for bringing up this point, we believe that applying HTI to hyperparameter optimisation with BO is a very interesting further potential use case. Standard BO trains a surrogate for a specific scalar objective. HTI trains a surrogate for the *output distribution*. This can therefore enable some kind of more flexible BO, where the user can change the objective function (e.g., from "maximize reward" to "maximize reward subject to safety constraints") post-hoc without needing to train a new surrogate model. We have expanded upon this in Section 3 and Appendix A. Nevertheless, we see the primary use case of HTI as allowing efficient inference-time adaptation of network behaviour. We believe this is a compelling enough use case for us to introduce this method, and leave extensive further use case experiemnts, such as with BO, to future works.
>
> **Update:** We have **expanded our discussion on BO** in **Section 3 and Appendix A**.
>
> ---
>
> Thank you for your time and effort. We hope that the we have made the practical potential of HTI more clear, and the new experiment provides further insights that the HTI framing allows. We are more than happy to engage with any further comments you may have.

---

> > ### Comment · Reviewer_rFXY · 2025-11-25
> >
> > I appreciate the author's work in addressing my concerns and explaining their changes. I feel that they have adequately addressed my concerns and edited the text to make scalability and practicality clearer. This is a novel and interesting contribution to the literature, and I am inclined to raise my score accordingly.

---

> > > ### Author Response · Authors · 2025-11-25
> > >
> > > Thank you for your time and effort throughout the review period, your comments were very helpful in improving our manuscript. We are very glad that we have addressed your concerns and we appreciate the positive assessment of our work!
> > >
> > > We did want to **note** that, as of now, we do not yet see an updated score from you on our end. If there is anything further you would like us to clarify or revise, we would be very happy to do so.

---

### Official Review · Reviewer_9CGQ · 2025-10-31

**Soundness:** 3
**Presentation:** 2
**Contribution:** 3
**Rating:** 4
**Confidence:** 3

**Summary:**

The paper defines Hyperparameter Trajectory Inference (HTI): given a neural model whose behaviour changes with a continuous hyperparameter, learn a surrogate that interpolates the model’s output distributions between a few trained anchor values. To do this, the authors extend neural Lagrangian optimal transport to the conditional setting: they learn, from sparse observations, a conditional Lagrangian and amortised CLOT maps/geodesics so that inferred paths stay low-action and remain in dense regions. They validate this on a 2D toy task, on two RL surrogates (cancer therapy, Reacher), and on quantile regression, where their full model (learned metric + density potential) is consistently better than ablations and the CFM/direct surrogates.

**Strengths:**

- HTI is a useful abstraction: many hyperparameters in practice (reward weights, discount factors, quantile levels, robustness coefficients) define families of policies or predictors, but current practice trains only a few points on that curve. Framing this as conditional trajectory inference and tying it to OT provides a principled way to discuss “hyperparameter-induced dynamics.”
- The authors do not simply make their approach learn pairwise conditional OT maps. It learns the cost itself via a conditional Lagrangian, with two inductive biases, and the Lagrangian is the device that lets them force the inferred paths to stay plausible when data are sparse.
- The experiments section seems to confirm that the approach enables learning from a few, sparse observations...

**Weaknesses:**

- ... but all (few) real evaluation tasks are fairly low-dimensional on the output side, forecasts with short horizons. The method is sold as applicable to “complex and higher-dimensional geometries,” but the experiments do not necessarily convey this statement.
- The RL scenarios considered are in fact well-behaved; the parameter $\lambda$ yields a linear combination of "objectives" (e.g., main reward+penalty/cost). In such cases, it is widely known that the resulting trade-offs from tuning $\lambda$ are well-behaved (see [1]).
- The introduction of core mathematical notions (conditional OT, the c-transform, and the Lagrangian formulation) is difficult to follow for readers not already fluent in OT theory and Lagrangian. For instance, the paper directly presents the dual form with the c-transform (eqs. 2-3) and the action integral (eq. 5) without first motivating them intuitively or connecting them to the simpler Kantorovich dual with explicit constraints. As a result, the rationale for moving from standard OT to a Lagrangian formulation, and why this path-based cost is required for CTI/HTI, remains implicit.
- The same issue carries over when explaining why the method can generalize from sparse observations: the text hints that the least-action and density biases provide this capability, but the link is scattered across sections 3 to 5 instead of being clearly stated when these concepts are first introduced. Overall, the exposition of the theoretical background is technically correct but conceptually opaque, making the narrative harder to follow than necessary.
- As acknowledged by the authors, a main limitation of the approach is that it only scales to a unique hyperparameter. In practice, many hyperparameters must be tuned. This fact is acknowledged only at the end of the paper; the limitation should be explicitly stated earlier in the text.
- Since one of the key selling points is the capability to learn from sparse observations, the evaluation would benefit from evaluating the quality of the predictions with respect to the sparsity of the observations (add or remove anchors and analyze the impact).

[1] Roxana Radulescu, Patrick Mannion, Diederik M. Roijers, Ann Nowé: Multi-objective multi-agent decision making: a utility-based analysis and survey. Auton. Agents Multi Agent Syst. 34(1): 10 (2020)

**Questions:**

- Could you explain why exactly you need to introduce the $c$-transform instead of the standard dual formulation?
- Concerning the move from classical COT to Lagrangian COT, could you articulate more clearly why a path-based cost is needed for CTI/HTI and whether using only the kinetic term with a learned metric (no potential) would already be enough for sparsity?
- How does Equation 12 help in evaluating Equation 11 faster? As you train $T\_{\theta}\_{T, k}$ via $\mathcal{L}\_{map}$, do you explicitly need to minimize Equation 11?
- Would you expect to obtain so clean results on RL experiments if the reward scalarization function were non-linear?

---

> ### Author Response · Authors · 2025-11-23
> **Part 1/2**
>
> Thank you for your thoughtful comments and suggestions, which have helped us significantly improve our manuscript. Please see the revised PDF, with changes in $\textcolor{blue}{\text{blue}}$. We also directly respond to your specific points below.
>
> ---
> **(A) Dual OT formulation**
>
> Thank you for raising this point, and helping us improve the clarity of our writing. We have now rearranged our preliminaries section, to first introduce the problem of HTI, and then the relevant mathematical machinery that allows us to address this.
>
> The primal COT problem (Eq. 1) is generally intractable, and cannot be easily neurally estimated as it requires modelling the high-dimensional joint coupling $\pi$. This often naturally leads to the use of the equivalent dual formulation (Eq. 2), which optimises two potential functions $f$ and $g$ subject to the constraint $f(x) + g(y) \le c(x,y)$ for all pairs $(x,y)$. With neural instantiations of these potentials, enforcing this constraint strictly across continuous high-dimensional spaces is difficult [1].
>
> The $c$-transform COT formulation (Eq. 3) serves as a constructive solution to this. By defining $f(x) = g^c(x) := \inf_y \{c(x,y) - g(y)\}$, the pair $(g^c, g)$ satisfies the inequality constraint optimally by construction. This converts the constrained optimisation problem into an unconstrained maximisation problem over a single potential $g$, which is significantly more tractable for neural network optimisation. This approach follows recent advances in Neural OT [2,3,4,5].
>
>
> **Update:** We have **revised Section 2** to clearly **motivate the progression from the standard COT formulation to the $c$-transform formulation**, explaining how this aids neural optimisation, and follows recent literature.
>
> ---
> **(B) Path-based OT cost**
>
> To see why we need a path-based OT cost, let's consider first the standard Euclidean OT cost, where $c(x,y) = ||x-y||^2$. This cost function is purely based on endpoints, because it implicitly assumes mass moves most efficiently along straight lines. In HTI, however, we hope to model the output distribution of NNs, which may evolve in more complex ways, potentially along non-Euclidean manifolds. If we used a fixed Euclidean cost, the inferred trajectory at intermediate $\lambda$ would be a linear interpolation between the source and mapped points, which ignores this potential complexity.
>
> To infer better intermediate distributions from sparse data, we look to impose inductive biases via the cost function. A Lagrangian action cost allows us to define the cost based on the *path* taken, not just the endpoints. This enables two specific biases:
> 1.  **Least action:** A learned metric encourages paths to reflect the geometry of the data. By assuming that observed data display movement in an manner that is efficient on the ambient manifold, we want to encourage inferred paths to mimic this movement.
> 2.  **Dense traversal:** The potential energy term $\hat{\mathcal{U}}$ penalizes paths that cross low-density regions (far from where data has been observed).
>
> These biases can only be incorporated via a **path-based cost function**, where the cost depends on all intermediate points that mass moves between two endpoints. This is what Lagrangian OT allows us to define, using the integral of the Lagrangian along the trajectory as path action.
>
> To address your specific question on whether the kinetic term alone ($K_{\theta}$) suffices, we can look to our experimental results in Section 5. Looking specifically at the $K_{\theta}$ ablation (which is precisely our method but using only the inductive bias of least-action, via the kinetic energy term) in **Tables 1, 2, 3, 4, and 6**, we see that this generally improves upon purely Euclidean COT (the $K_I$ ablation), but consistently performs worse than the full method ($K_\theta - \hat{\mathcal{U}}$). **Both** of the kinetic and potential terms individually help reconstruct realistic paths from sparse observations.
>
> **Update:** We have **expanded the motivation for the Lagrangian formulation** in **Section 2**, explicitly linking the need for valid intermediate samples to the requirement for path-based costs.

---

> ### Author Response · Authors · 2025-11-23
> **Part 2/2**
>
> **(C) Amortisation of the CLOT map**
>
> Calculating the $c$-transform $g^c(y|x)$ to evaluate the main objective (now Eq. 12) requires solving nested optimisation problems. If solved from scratch (e.g., via L-BFGS) at every training step, this can be prohibitively slow. The amortisation network $T_{\theta_{T,k}}$ allows us to predict this optimal map in an efficient manner like so:
>
>  - We use the output of $T_{\theta_{T,k}}$ to **warm-start** the $c$-transform optimisation. In doing so, we only need a few optimisation steps (e.g., we use 3 L-BFGS steps in our `Cancer` and `Reacher` experiments) during training, making Eq. 12 computationally feasible.
>  - The optimised value is used in Eq. 12 **and** as the training target for Eq. 13, improving the amortised map's performance each iteration.
>
> $T_{\theta_{T,k}}$ confers additional benefits at inference time, allowing us to sample from the surrogate model by predicting CLOT maps with a single forward pass, instead of having to solve any optimisation problem at test time.
>
> **Update:** We have **clarified the training procedure in Section 4.2**, and **moved Algorithm 1** to the main body of the paper, to improve the clarity of the amortisation procedure.
>
> ---
> **(D) More complex environments**
>
> Thanks for this point, and the relevant reference, as we do indeed hope to evaluate our method in complex settings. We now have added an experiment that considers performance for RL reward weighting when the reward scalarization is **non-linear**. To do so, we introduced a non-linear hinge penalty experiment in the `Cancer` environment, inspired by Eq. 6 of (Rǎdulescu et al., 2020).
>
> In this setup, the NK cell penalty is only applied if the cell count changes beyond a specific threshold. This creates a non-linear reward scalarization, which could result in more difficult to model hyperparameter dynamics. The surrogate reward results are presented below, and in Section 5.2.3. Our full method ($K_\theta - \hat{U}$) continues to outperform baselines, demonstrating that HTI can model more complex, non-linear dynamics
>
> | Method | Reward ($\uparrow$) |
> | :--- | :---: |
> | $\mathcal{K}_I$ | $42.84$ $(5.86)$ |
> | $\mathcal{K}_\theta$ | $45.83$ $(12.73)$ |
> | Direct | $49.50$ $(17.90)$ |
> | CFM | $69.70$ $(7.73)$ |
> | $\mathcal{K}_I - \hat{\mathcal{U}}$ | $94.93$ $(5.83)$ |
> | $\mathcal{K}_\theta - \hat{\mathcal{U}}$ | $101.80$ $(7.93)$ |
>
> **Update:** We have **included these new results in Section 5.2.3** to demonstrate good performance in RL settings with **non-linear reward scalarization**.
>
> ---
> **(E) Singular hyperparameter limitation**
>
> We agree that this is a limitation of the current method. We now explicitly state earlier (in Sections 1 and 2) that our method focuses on a **single** hyperparameter.
>
> **Update:** We now **explicitly state that we focus on single hyperparameter-dynamics** in **Sections 1 and 2**.
>
> ---
> **(F) Sparsity investigation**
>
> We agree that an explicit study of how sparsity affects results will improve the manuscript. To analyse inference quality with respect to the sparsity of observations, we conducted a new ablation in the `Cancer` and `Reacher` environments, training HTI surrogates with data from varying numbers of anchor policies. We display the results in Figure 3, in Section 5.3. We see that, in both environments, the performance gap between methods is negligible in the dense regime, where interpolation is trivial, and this widens as sparsity increases. Our method degrades the least as interpolation becomes more difficult, outperforming all baselines in sparse settings, confirming that our inductive biases are especially effective when sparsity makes interpolation difficult.
>
>
> **Update:** We have **added this new sparsity analysis** to **Section 5.3**.
>
> ---
> Thank you for your time and effort. We hope that the refined mathematical preliminaries, clarified training procedure, and the new experimental evidence address your concerns. We are more than happy to engage with any further comments you may have.
>
> ---
>
> [1] Vivien Seguy, Bharath Bhushan Damodaran, Rémi Flamary, Nicolas Courty, Antoine Rolet, and
> Mathieu Blondel. Large-scale optimal transport and mapping estimation. arXiv preprint
> arXiv:1711.02283, 2017.
>
> [2] Amirhossein Taghvaei and Amin Jalali. 2-wasserstein approximation via restricted convex potentials
> with application to improved training for gans. arXiv preprint arXiv:1902.07197, 2019.
>
> [3] Ashok Makkuva, Amirhossein Taghvaei, Sewoong Oh, and Jason Lee. Optimal transport mapping via
> input convex neural networks. In International Conference on Machine Learning, pp. 6672–6681.
> PMLR, 2020.
>
> [4] Brandon Amos. arXiv:2210.12153, 2022. On amortizing convex conjugates for optimal transport. arXiv preprint
>
> [5] Aram-Alexandre Pooladian, Carles Domingo-Enrich, Ricky Tian Qi Chen, and Brandon Amos.
> Neural optimal transport with lagrangian costs. In Proceedings of the Fortieth Conference on
> Uncertainty in Artificial Intelligence, 2024.

---

> > ### Comment · Reviewer_9CGQ · 2025-11-27
> >
> > Thank you for your strong rebuttal.
> > I think the presentation of the background is way clearer now. I also like the updated presentation of the OT notions.
> > I appreciate the fact that you included new experiments, especially for point (D). Just a small remark: you should clearly refer to the equations behind the scalarization function from the Appendix in the main text. I originally thought you simply did not provide them. It is provided, but you do not mention it explicitly (I note a broad statement at the very beginning of section 5).
> >
> > I'm still finding that the current experiments are fairly low-dimensional, but I note that the experimental section has been substantially strengthened during the rebuttal period, with a non-linear RL scalarisation, a sparsity study, and a direct comparison to NLOT.
> >
> > As you mostly addressed my concerns, I'm updating my score accordingly.

---

> > > ### Author Response · Authors · 2025-11-28
> > >
> > > Thank you for your continued engagement with our work, your comments have been instrumental in improving our manuscript. We have now added an explicit reference in Section 5.2.3 to the non-linear reward scalarization definition in the Appendix, as suggested.
> > >
> > > We agree that evaluating HTI methods in more complex and larger settings is an important direction for future work. Our comments in the paper about “higher-dimensional” settings are primarily intended to highlight that our **metric parametrisation**, unlike the related NLOT approach, does not impose a strict limit of 2D ambient spaces, and can be extended, as in Section 5.5.
> > >
> > > We are glad that we have addressed your concerns, and we are grateful for the score increase!

---

### Author Response · Authors · 2025-11-23

We thank the reviewers for their time and constructive feedback. We are encouraged that the reviewers found our problem setting novel and impactful, and our method elegant and effective. Based on your suggestions, we have significantly revised the manuscript to improve clarity, flow, and experimental rigor.

Below is a summary of the major changes and where to find them in the updated manuscript (changes are marked in the PDF in $\textcolor{blue}{\text{blue}}$):

**1. Structural Reorganization**
*   **Problem Definition & Preliminaries (Section 2):** We have restructured the paper to introduce the problem of **Hyperparameter Trajectory Inference** at the beginning of Section 2. We then revised the mathematical preliminaries to explicitly motivate the progression from standard Conditional OT to the $c$-transform formulation and Lagrangian path-based costs, better linking our mathematical framework to the HTI problem statement.
*   **Training (Section 4):** We have moved **Algorithm 1** from the appendix to the main body (Section 4) to clarify the training procedure.

**2. New Experiments and Analyses**
*   **Sparsity Investigation (Section 5.3):** We added a new analysis evaluating how hyperparamter trajectory reconstruction quality degrades as the number of anchor policies decreases. We see that our method degrades the least in sparse, difficult settings, while all methods perform similarly in dense, easy settings.
*   **Non-Linear Reward Scalarization (Section 5.2.3):** We added an experiment using a non-linear reward scalarization, with a hinge penalty, in the `Cancer` environment. This demonstrates that our method can effectively, and better than baselines, model hyperparameter dynamics even when the reward scalarization is non-linear and complex.

**3. Additional Baselines and Comparisons**
*   **Comparison with NLOT (Appendix E):** We added a direct comparison against a conditional version of the related Neural Lagrangian OT (NLOT) method from Pooladian et al. (2024), which our method significantly outperforms.

**4. Expanded Discussions**
*   **Bayesian Optimization (Section 3 & Appendix A):** We expanded the discussion on further use cases for HTI, beyond inference-time speed ups. We specifically focus on how HTI can complement standard Bayesian Optimization by enabling post-hoc objective changes.

We believe this has been a very productive review period, and that these changes have strengthened the manuscript considerably. We hope that the reviewers' concerns have been addressed.

---

### Author Response · Authors · 2025-12-03
**Summary for the Area Chair**

Dear Area Chair,

Thank you for your oversight of the review of our paper, particularly given the recent increase in your workload. To help with your assessment, here is a brief summary of our discussions with the reviewers. **Please note that, in our discussions, we addressed the concerns of each reviewer, which they all explicitly acknowledged**. As a result, we had **Reviewer Ei6o** change their score from a **2 to an 8**, **Reviewer 9CGQ** change their score from a **4 to an 8**, and **Reviewer rFXY** state that they were **"inclined to raise [their] score" from a 6.**

We summarise the changes that lead to these scores increases below.

| Change | Summary | Reviewer Addressed |
| :--- | :---: | :---: |
| **Extended Preliminaries** | We now introduce the problem of HTI at the beginning of Section 2. We also expand on the mathematical preliminaries to better motivate the progression from the primal optimal transport problem to the $c$-transform formulation, with Lagrangian path-based costs, better linking our mathematical framework to the HTI problem statement.| 9CGQ, Ei6o |
| **Method** |  We have moved Algorithm 1 from the appendix to Section 4, to improve readability and clarify the training procedure.| 9CGQ, Ei6o |
| **Non-Linear RL Experiment** | We have added a new experiment, investigating HTI performance with a non-linear reward scalarization in the `Cancer` environment. Our method can effectively, and better than baselines, model hyperparameter dynamics in this more complex setting.| 9CGQ |
| **Sparsity Experiment** | We added a new experiment evaluating how HTI performance degrades as the number of anchor policies decreases in the `Cancer` and `Reacher` environments. We see that our method especially outperforms in difficult, sparse settings. | 9CGQ, rFXY |
| **Additional Baseline** | We added a direct comparison against a conditional version of the related Neural Lagrangian OT (NLOT) method from Pooladian et al. (2024), which our method significantly outperforms. | rFXY, Ei6o |
| **Expanded Practical Discussion** | We expanded on further use cases for HTI, beyond inference-time speed ups, highlighting how HTI can complement Bayesian Optimization by enabling post-hoc objective changes. | rFXY |

We are very pleased with the positive assessment that each reviewer now has, because of the above changes. Please see the full discussion threads, with responses from each reviewer, for further evidence of this. We hope that you agree that we have addressed the concerns of the reviewers, and significantly improved the quality of our manuscript. Thank you again for your time and effort.

---

### Meta-Review · Area_Chair_5XDc · 2026-01-07

**Summary:**

The paper introduces a new task Hyperparameter Trajectory Inference (HTI) and proposes a framework based on conditional Lagrangian optimal transport to learn how neural network’s behaviours vary depending on hyperparameters, enabling accurate surrogate inference at unseen hyperparameter settings.

The initial ratings were mixed and slightly lower than the acceptance threshold. The reviewers raised several concerns:

1. Clarity and structure of writing, including preliminaries, the HTI problem statement, and several presentation suggestions.
2. Experiments in more complex settings, including nonlinear and sparse settings.
3. Further discussion about the practicality of the proposed method.

**Reviewer Concerns:**

The authors effectively addressed most concerns as

1. Most suggestions were incorporated in the revised version and the authors highlighted the updated parts in blue, including the rearrangement of the main algorithm from the appendix to the main paper.
2. The authors provided additional experiments under more complex settings, demonstrating the effectiveness of the proposed method. Also, the authors added a direct comparison with an additional baseline.
3. The authors added a discussion about practicality and the relationship with Bayesian optimization, which is often used for automatic hyperparameter tuning in the context of AutoML.

**Reviewer Scores:**

As the reviewers acknowledged, the authors effectively addressed most major concerns and the two reviewers explicitly commented that they raised the ratings as 4 to 6, 2 to 8. Also, one reviewer with a rating of 6 indicated an intention to increase the rating. Overall, the rebuttal significantly improved this work and the ratings would lead to a clear accept if the full discussion period is given. Therefore, acceptance is recommended.

---

### Decision · Program_Chairs · 2026-01-26

Accept (Oral)